# Capacitive in-sensor tactile computing

Yan Chen[1,2,3], Jie Cao ●[1], Jie Qiu ●[1,2,3], Dongzi Yang[1,2], Mengyang Liu[1], Mengru Zhang[1,2], Chenyang Li[1,2,3], Zhongyuan Wu[4], Jie Yu[1], Xumeng Zhang ●[1], Xianzhe Chen ●[1], Zhangcheng Huang ●[1], Enming Song ●[4], Ming Wang ●[1,3] ✉, Qi Liu ●[1,3] & Ming Liu[1,3] ✉

Real-time sensing and processing of tactile information are essential to enhance the capability of artificial electronic skins (e-skins), enabling unprecedented intelligent applications in tactile exploration and object manipulation. However, conventional tactile e-skin systems typically execute redundant data transfer and conversion for decision making due to their physical separation between sensors and processing units, leading to high transmission latency and power consumption. Here, we report an in-sensor tactile computing system based on a flexible capacitive pressure sensor array. This system utilizes multiple connected sensor networks to execute in-situ analog multiplication and accumulation operations, achieving both tactile sensing and computing functionalities. We experimentally implemented the in-sensor tactile computing system for low-level tactile sensory processing tasks including noise reduction and edge detection. The consumed power for single sensing-computing operation is over 22 times lower than that of a conventional mixed electronic system. These results demonstrate that our capacitive in-sensor computing system paves a promising way for power-constrained applications such as robotics and human-machine interfaces.

Tactile electronic skins (e-skins) can enhance the human and robot to understand their surrounding environment, enabling unprecedented applications in tactile exploration and object manipulation, such as neuroprosthetics[1–3], robotics[4–7], and human-machine interfaces[8–10]. Such exploration and manipulation typically rely on the rapid detection and processing of unstructured, redundant and analog tactile signals during interactions with target objects[5,11], including pressure, strain, and temperature information. However, in a conventional tactile e-skin system, analog tactile signals are captured by tactile sensors, converted into digital format via analogue-to-digital conversion circuits and subsequently transmitted to external processing units for computation[12,13]. This physical separation between sensors and processing units brings large amounts of data conversion and transfer, resulting in high transmission latency and power consumption.

Recent advances in near-sensor and in-sensor computing paradigms improve the processing efficiency of the sensing and processing system via reducing or even eliminating the interface between sensors and processing units[13–25]. In the near-sensor computing paradigm, data computation is performed beside the sensors[13,26], which reduces the transfer of redundant data but still cannot escape the dilemma of the physical separation of sensors and processing units. In contrast, in-sensor computing paradigm utilizes individual self-adaptive sensors[27], multiple connected sensors[13], or novel device structures[28,29] to directly sense and simultaneously process sensory information, providing a more attractive solution with the relatively complete elimination of data conversion and transfer in the system. Especially, the in-sensor computing at the array level can implement analog multiplication and accumulation (MAC) operations that are a prerequisite to implement an artificial neural network[17,30]. The in-sensor MAC computing

[1]State Key Laboratory of Integrated Chips and Systems, Frontier Institute of Chip and System, Zhangjiang Fudan International Innovation Center, Fudan University, Shanghai, China. [2]College of Integrated Circuits and Micro-Nano Electronics, Fudan University, Shanghai, China. [3]Zhangjiang Laboratory, Shanghai, China. [4]Shanghai Frontiers Science Research Base of Intelligent Optoelectronics and Perception, Institute of Optoelectronics, Fudan University, Shanghai, China. ✉e-mail: wang_ming@fudan.edu.cn; liuming@fudan.edu.cn

paradigm has been well demonstrated in optoelectronic devices and arrays for visual sensory processing[31–33]. However, the implementation of in-sensor MAC computing for tactile stimuli has not yet reported (Supplementary Table 1)[28,29,34,35], primarily due to the absence of direct computing capability in individual tactile sensors. Among various tactile sensing technologies, capacitive-type sensors offer the advantages of high sensitivity, excellent stability, fast response time, and lower power consumption[36,37]. Although interconnected capacitive sensor arrays show potential to sense and process tactile stimuli simultaneously, implementing in-sensor MAC operations for tactile stimuli remains elusive.

In this work, we report an in-sensor tactile computing system based on a flexible capacitive pressure sensor array. This system utilizes multiple connected capacitive sensor networks to implement the tactile MAC operation with the assistance of electrical switches, realizing real-time sensing and simultaneous computing of tactile stimuli. The capacitive sensor array is fabricated by the stacked structure with a microstructured polyvinyl alcohol/phosphoric acid (PVA/$H_3PO_4$) sensing layer and stretchable gold (Au) electrodes, exhibiting a high capacitance-pressure sensitivity of $0.36\,nF\cdot kPa^{-1}$. We experimentally illustrate the in-sensor tactile computing system for low-level tactile sensory processing tasks, including noise reduction and edge detection. The maximum power consumption of our system is only $493\,\mu W$ and $492\,\mu W$ for tactile noise reduction and edge detection, respectively, exhibiting over 22 times lower than a conventional mixed

electronic system. These results demonstrate that our capacitive in-sensor computing system is promising for power-constrained applications in tactile exploration and object manipulation, including robotics and human-machine interfaces.

## Results

### Principle of capacitive in-sensor tactile computing

Figure 1a illustrates the schematic of capacitive in-sensor tactile computing system built upon a capacitive pressure sensor array and electrical switches. The sensor array consists of top and bottom substrate layers, top and bottom electrode layers and a sensing layer. In the array, the top electrode of each sensor pixel is connected to two individual electrical switches (T1 and T2), and the bottom electrodes of all sensor pixels are grounded. Multiple pressure sensor pixels in the array are interconnected via their corresponding T2 switches, forming a capacitive in-sensor tactile computing subregion (dotted box in Fig. 1a). An additional electrical component such as a fixed capacitor ($C_O$) is employed to read out the calculated result of in-sensor tactile computing subregion.

The operation principle of capacitive in-sensor tactile computing is illustrated as follows (Fig. 1b). For each pressure sensor pixel ($C_1$ to $C_n$), the sensor generates a deformation and changes its capacitance value to respond to an external tactile stimulus. When T1 switches turn on and T2 switches turn off, electric charges will flow into the top electrode of pressure sensor pixels, driven by a series of voltage biases

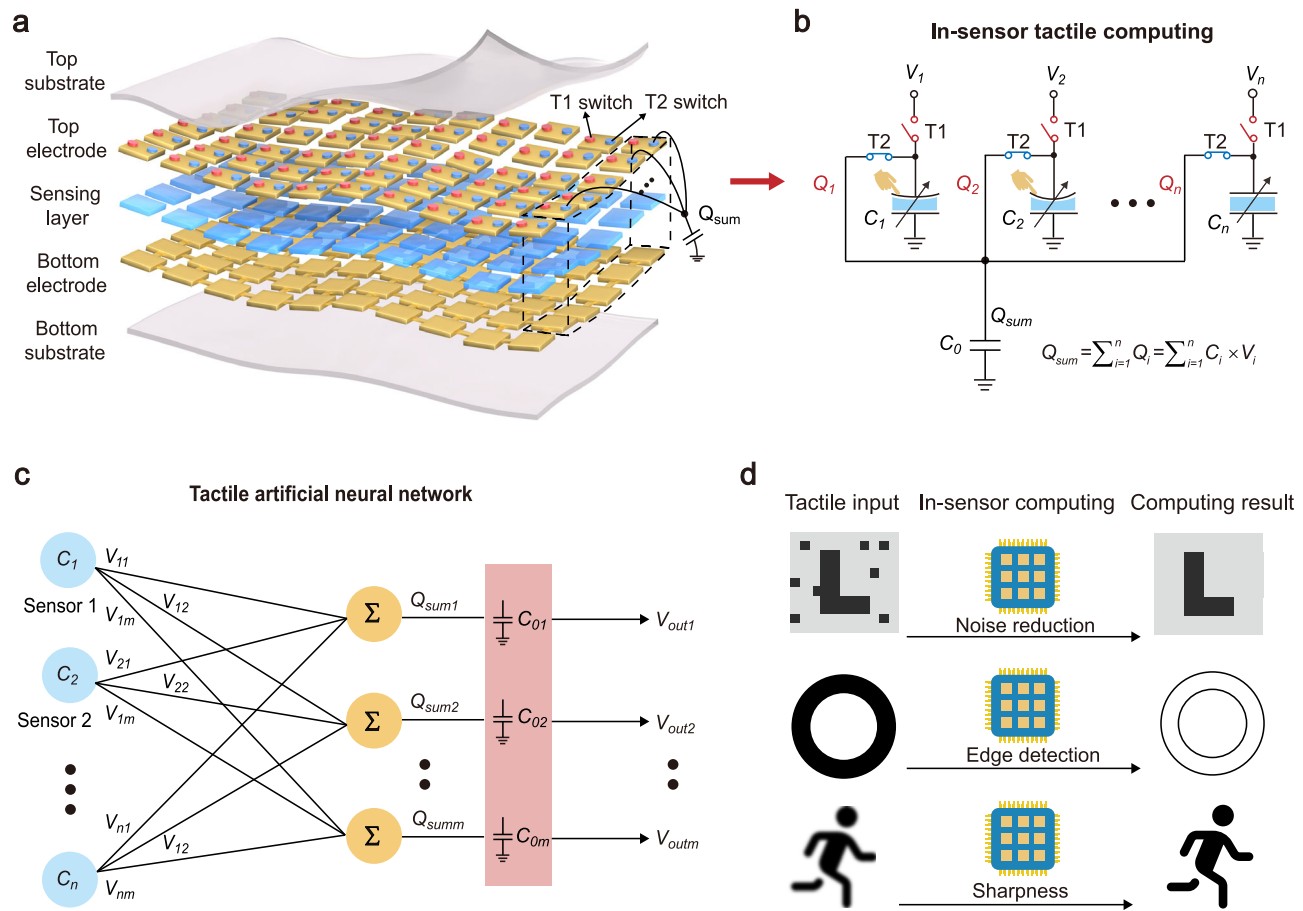

**Fig. 1 | Capacitive in-sensor tactile computing system. a** Schematic of the capacitive in-sensor tactile computing system based on a flexible capacitive tactile sensor array and electrical switches. **b** Physical processes of realizing the capacitive in-sensor tactile computing. Multiple connected capacitive tactile sensors in the array (dotted box shown in (**a**)) form an in-sensor tactile computing subregion, implementing in-situ multiplication and accumulation (MAC) operation. **c** Tactile artificial neural network constructed by the input capacitive pressure sensor vector **C** and predefined voltage matrix **V**. **d** Capacitive in-sensor tactile computing for low-level sensory computation tasks, such as noise reduction, edge extraction, and sharpness.

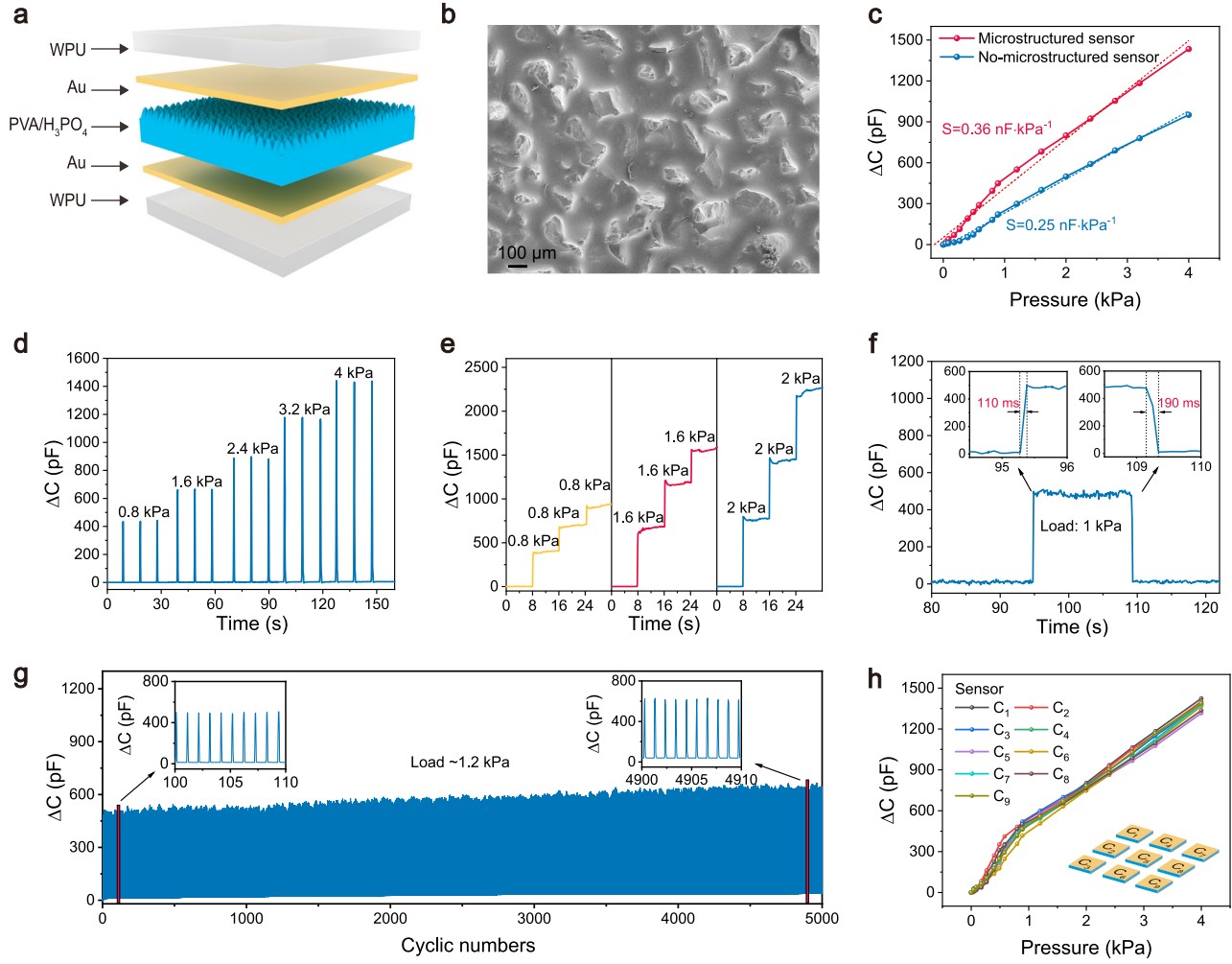

**Fig. 2 | Characterization of the flexible capacitive pressure sensor array.**
**a** Structure diagram of the capacitive pressure sensor unit in the array. **b** Top-view scan electron microscopy (SEM) image of the fabricated PVA/H₃PO₄ film, showing a rough microstructured surface. **c** Capacitance-pressure response of the capacitive sensors with and without the microstructure. **d** Changes in capacitance value of the pressure sensor under five loading and unloading pressure stimuli of 0.8, 1.6, 2.4, 3.2, and 4 kPa. **e** Changes in capacitance value of the pressure sensor under the sequential loading pressure stimuli of 0.8, 1.6, and 2 kPa over eight seconds. **f** Dynamic capacitance-pressure response. The response time and recovery time are 110 ms and 190 ms, respectively. **g** Cyclic loading and unloading test under a maximum pressure stimulus of 1.2 kPa. **h** Capacitance-pressure response of all nine sensor units in the array. The applied pressure ranges from 0 to 4 kPa.

($V_1$ to $V_n$) applied on the T1 switches. As a result, each capacitive pressure sensor accumulates a certain amount of charges ($Q_1$ to $Q_n$) depending on the value of the applied tactile stimulus and its corresponding voltage bias, referred to as a charging process. The accumulated charges stored on each capacitive pressure sensor can be expressed as $Q_i = C_i \times V_i$, where $C_i$ and $V_i$ denote its capacitance value and voltage bias at the $i_{th}$ sensor pixel ($i = 1,2...n$), respectively. Subsequently, as T1 switches turn off and T2 switches turn on, the individual charges ($Q_i$) stored on each pressure sensor pixel will converge to form a summed charge ($Q_{sum}$) via a charge sharing process. The summed $Q_{sum}$ can be read via the peak voltage ($V_{out}$) across a fixed parallel capacitor $C_O$ (Supplementary Note 1). The summed $Q_{sum}$ represents the calculated result of capacitive in-sensor tactile computing, given by:

$$Q_{sum} = \sum_{i=1}^{n} Q_i = \sum_{i=1}^{n} C_i \times V_i, i = 1, 2 \ldots n \qquad (1)$$

As a result, a tactile MAC dot product operation ($Q_{sum} = \mathbf{CV}$) can be performed in the charge domain based on the two physical charging and sharing processes, where $\mathbf{C} = (C_1, C_2, ..., C_n)$ denotes the capacitive pressure sensor vector, $\mathbf{V} = (V_1, V_2, ..., V_n)$ denotes the input

voltage vector, and $Q_{sum}$ represents the calculated output. Multiple MAC dot product operations can be executed in parallel within the capacitive pressure sensor array (Supplementary Fig. 1). Multiple capacitive pressure sensors can physically implement a tactile artificial neural network (Fig. 1c). Combined with the predefined voltage matrix **V** (reconfigured as specific positive or negative elements), the capacitive sensor array **C** can be used to perform various sensory computation tasks, such as noise reduction, edge extraction and sharpness (Fig. 1d). As a result, the capacitive pressure sensor array can detect tactile stimuli in real time and simultaneously process the monitoring tactile information, offering an in-sensor tactile computing paradigm for advanced e-skin applications.

## Fabrication and characterization of the capacitive sensor array

We experimentally fabricated a flexible capacitive pressure sensor array to illustrate the validity of our proposed capacitive in-sensor computing system. The pressure sensor pixel in the array shows a multi-layer stacked structure consisting of waterborne polyurethane (WPU) top substrate, Au top electrode, PVA/H₃PO₄ sensing layer, Au bottom electrode, and WPU bottom substrate (Fig. 2a). The fabrication processes of the capacitive pressure sensor array are illustrated in detail in Methods and Supplementary Fig. 2. Briefly, Au layers were

directly deposited on a prefabricated stretchable WPU substrate as the top and bottom electrodes using the shadow mask process[38]. Ionic elastomeric PVA/H$_3$PO$_4$ film was selected as the capacitive sensing layer because of its high dielectric constant and ease of mass fabrication[39]. The elastomeric film was fabricated by drop-coating the PVA/H$_3$PO$_4$ solution onto a predefined rough mold and curing to obtain a microstructured sensing layer (Fig. 2b). Finally, the patterned bottom electrode, microstructured sensing film, and patterned top electrode were vertically stacked to form a capacitive pressure sensor array (Supplementary Fig. 3).

We firstly characterized the capacitance-pressure response of the capacitive pressure sensor (Fig. 2c). The pressure sensor with the microstructured surface exhibits a high sensitivity of 0.36 nF•kPa$^{-1}$ in the pressure range of 0–4 kPa, which is larger than that of 0.25 nF•kPa$^{-1}$ for the pressure sensor without the microstructure surface. This improvement can be explained that the microstructure increases the compression ratio of the PVA/H$_3$PO$_4$ sensing layer under external pressure stimuli. Device-to-device uniformity was confirmed through testing of 30 sensors (Supplementary Fig. 4), showing excellent consistency. In addition to high sensitivity, the capacitive sensor needs to exhibit good stability and repeatability for high-quality tactile signal acquisition, which is critical to the in-sensor tactile computing. Figure 2d illustrates the real-time sensing responses of the capacitive pressure sensor under five loading and unloading pressure stimuli of 0.8, 1.6, 2.4, 3.2, and 4 kPa. For each cycle at the same stimulus, the sensor exhibits similar monitoring capacitance values, demonstrating its good stability and repeatability. When the sequential loading pressure stimuli of 0.8, 1.6, and 2 kPa are steadily applied over eight seconds, the sensor shows a distinct stepwise increase in the capacitance value for each kind of pressure stimulus (Fig. 2e), further verifying the stability of the capacitive pressure sensor.

To evaluate the response speed of the sensor, a pressure stimulus of 1 kPa is applied to the sensor for a period of time and then rapidly released to monitor the capacitance-pressure response (Fig. 2f). Enlarged pressure-capacitance sensing curves reveal that the sensor has a fast response and recovery time of 110 ms and 190 ms, respectively (Inset of Fig. 2f). Additionally, the sensor can undergo 5000 times of cyclic loading and unloading tests at a peak pressure stimulus of 1.2 kPa, exhibiting a small signal fluctuation (Fig. 2g). Furthermore, all capacitive pressure sensors in the array show similar capacitance-pressure response curves in the pressure range of 0–4 kPa (Fig. 2h), verifying an excellent device uniformity. These results demonstrate the microstructured capacitive pressure sensor shows high sensitivity, excellent reliability and good uniformity, guaranteeing the reliability of the capacitive in-sensor tactile computing system.

## In-sensor computing in the capacitive sensor array

To implement a specific sensory computation task, the flexible capacitive pressure sensor array can be configured as an in-sensor tactile computing kernel (Fig. 3a and Supplementary Fig. 5), which consists of nine adjacent capacitive pressure sensor pixels ($C_1$ to $C_9$) with their corresponding voltage biases ($V_1$ to $V_9$). The computing kernel can detect the tactile stimulus projected onto the array in real time and simultaneously generate an output voltage $V_{out}$ as the calculated result (when a fixed capacitor $C_O$ as the readout component).

According to the working mechanism illustrated in Fig. 1b, a whole process to physically implement the tactile MAC operation requires three phases ($\Phi_1$, $\Phi_2$, $\Phi_3$) for the capacitive in-sensor tactile computing kernel (Fig. 3b, c). The detailed operation flow of the kernel is illustrated in the Supplementary Note 1 and Supplementary Fig. 6. At the first $\Phi_1$ phase, a refresh operation is performed by turning on the T3 switch and turning off all T1 and T2 switches, ensuring the completely clear the charges possibly stored on the readout capacitor $C_O$. At the second $\Phi_2$ phase, nine T1 switches turn on and other switches turn off. All pressure sensors in the array will be directly connected to their corresponding predefined voltage biases ($V_1$ to $V_9$) and are subsequently charged, implementing nine multiplication operations in parallel ($Q_i = C_i \times V_i$). At the third $\Phi_3$ phase, the nine T1 switches simultaneously turn off and then the nine T2 switches turn on, the majority of charges stored on each sensor pixel will transfer onto the readout capacitor $C_O$ (Supplementary Note 1), implementing the accumulation operation. At the moment when the T2 switches change from on to off states, the potential value ($V_{out}$) on the readout capacitor $C_O$ is read out as the final calculated result.

Depending on the combinational configuration of the predefined voltage biases ($V_1$ to $V_9$), the in-sensor tactile computing kernel can execute various low-level tactile sensory processing tasks. For instance, when the voltage biases from $V_1$ to $V_9$ are predefined to an identical value, the kernel is configured as an averaging filter that can detect the tactile stimuli in real time and simultaneously remove noise signals. Figure 3d shows the calculated results of the in-sensor tactile computing kernel under different tactile stimulus patterns. Here, all voltage biases are 3.3 V, and the readout capacitor $C_O$ is 10 nF. When four sensor pixels in the kernel are applied with the pressure stimuli of 2 kPa, the calculated $V_{out}$ is about 1.084 V. These calculated results match the simulation results with a maximum deviation of 0.035 V (Supplementary Note 2), demonstrating that feasibility of the in-sensor tactile averaging computing kernel. It should be noted that the in-sensor tactile computing kernel generates an output voltage value for each operation, representing the calculated result of a tactile stimulus pattern (3 × 3 pixels). The interval time between two operations is set to 42 ms, which can be reduced to 1 ms or less for faster processing (Supplementary Fig. 7a). In addition, the turn-on time for all T1, T2, and T3 switches can be shortened to further improve speed (Supplementary Fig. 7b).

For a real object consisting of large-scale pixels, the in-sensor tactile computing kernel is required to perform a raster scanning operation to process all pixels (Fig. 4a). The raster scanning process is analogous to the implementation of a convolutional operation in the artificial neural network algorithm[32,40]. To illustrate this operation for noise reduction, we designed a bullet-shaped mold with stochastic noise dots as a detected object (Fig. 4a). The noise mold contains 17 × 17 pixels, which was fabricated by 3D printing technology (Supplementary Fig. 8). When the 3 × 3 in-sensor tactile averaging computing kernel is pressed onto the noise mold, the kernel can sense the local structural information (3 × 3 pixels) of the mold and process the current tactile stimulus pattern to generate an output voltage. The output voltage is regarded as a new pixel of the processed object, which is obtained by concurrently computing the structural information from the nine correlative pixels in the kernel. After the in-sensor tactile averaging computing kernel scanning the entire mold (17 × 17 pixels) with a stride of one, all pixels in the noise mold will be sensed and computed, forming a new pattern (15 × 15 pixels) after noise reduction. In the current implementation, our tactile computing kernel requires physical sliding along both x- and y-axes to process large-area tactile inputs. This limitation can be overcome by implementing a large-area sensor array with properly sequenced electrical switches (Supplementary Fig. 9 and Supplementary Note 3), eliminating mechanical movement and enhancing practical applicability. For future large-scale integration, transistor-based switching could be employed to replace discrete switches.

Figure 4b illustrates the calculated results of the in-sensor tactile averaging computing kernel under all tactile stimulus patterns in the noise bullet-shaped mold. The calculated $V_{out}$ values range from 0.36 V to 1.57 V as tactile stimulus pattern change from mode 0 to mode 9 (Supplementary Fig. 10). All calculated $V_{out}$ values using the in-sensor tactile averaging computing kernel are consistent with the simulated values (Supplementary Note 2 and Supplementary Table 2). After a raster scanning operation, two new patterns obtained from the physical kernel and simulation are shown in Supplementary Fig. 11,

**a**

**In-sensor tactile computing kernel**

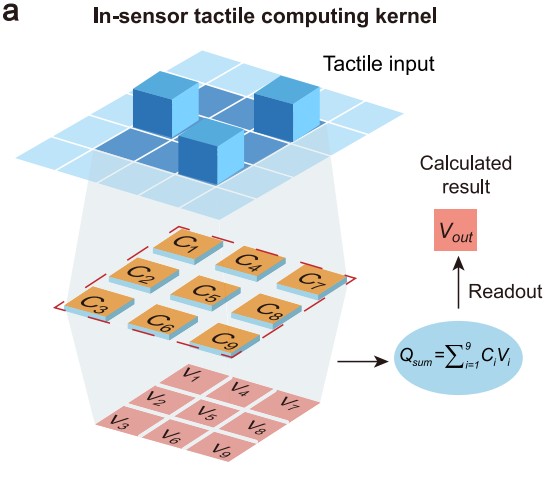

**b**

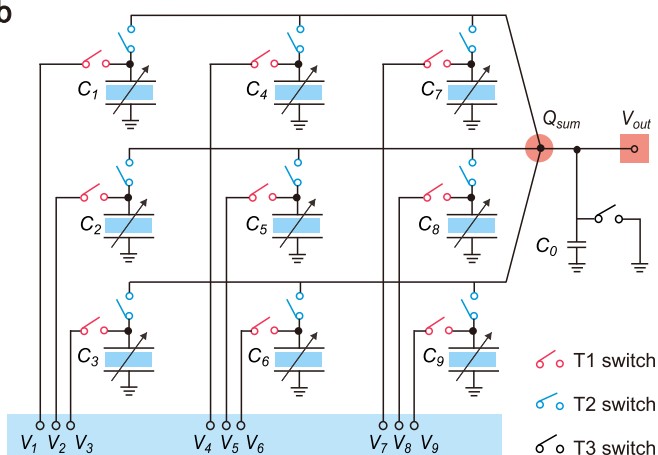

**c**

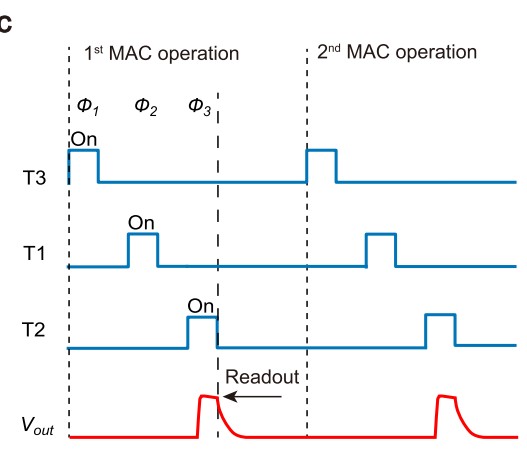

**d**

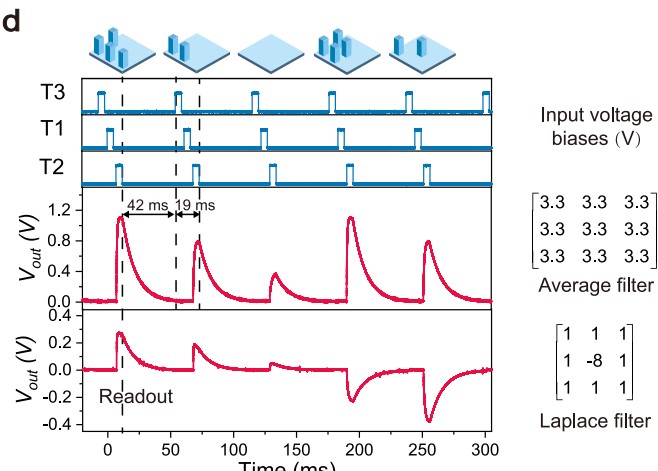

**Fig. 3 | In-sensor computing in the capacitive sensor array. a** Configuration of the capacitive in-sensor tactile computing kernel. The kernel consists of nine adjacent sensor pixels ($C_1$ to $C_9$) with corresponding voltage biases ($V_1$ to $V_9$), in which sensor $C_5$ is at the center of the array and is surrounded by other eight sensors. Blue pillar: pressure stimulus; none: no input. **b** Circuit diagram of the capacitive in-sensor computing kernel. **c** Timing diagram of the capacitive in-sensor computing kernel to implement the tactile MAC operation. A whole tactile MAC operation requires three phases ($\Phi_1$, $\Phi_2$, $\Phi_3$). The calculated result ($V_{out}$) is read out at the moment when the T2 switches turn off. **d** Calculated result of the capacitive in-sensor computing kernel under different tactile stimulus patterns. Blue pillar: 2 kPa; none: 0 kPa.

respectively. After the binarization, the two new patterns show a highly identical (Fig. 4c, d), demonstrating the in-sensor tactile computing for e-skin perception is possible and feasible.

In addition, we could configure the voltage bias $V_S$ in the center of the kernel to be a negative value of −8 V, and the surrounding voltage biases to be a positive value of 1 V. Such configuration including both positive and negative values forms a classical Laplacian filter (Fig. 3d), which is capable of executing edge detection. When four surrounding sensor pixels in the array are applied by the pressure stimuli of 2 kPa, the calculated $V_{out}$ is 0.24 V (Fig. 3d). In contrast, when the center pixel and three surrounding sensor pixels are applied by the pressure stimuli of 2 kPa, the calculated $V_{out}$ is −0.22 V. Figure 4e illustrates the calculated $V_{out}$ values of all tactile stimulus patterns using the in-sensor tactile Laplacian computing kernel. Again, these calculated values are highly consistent with the simulated values (Supplementary Fig. 12 and Supplementary Table 3), further demonstrating the feasibility of the in-sensor tactile computing kernel. Similar to the noise reduction process, we could use a raster scanning operation to obtain the whole contour information of a detected object. The edge detection results for a cross-shaped object (Supplementary Fig. 13) processed by the in-sensor tactile computing kernel also match with the simulated results

(Fig. 4f, g). These results demonstrate that our capacitive in-sensor tactile computing system holds a promising application for advanced e-skins.

### Capacitive in-sensor tactile computing for analog stimuli

In most scenarios, a real detected object usually has an irregular surface with the complex texture structure[41,42], which conveys analog tactile stimulus inputs into the flexible capacitive pressure sensor array. For instance, when a cat uses its claw to contact the sensor array, tactile inputs with multiple stimulus intensities will generate due to different regions of the claw with varied heights and materials properties (Fig. 5a). To demonstrate the capacitive in-sensor tactile computing system with the capability to process analog tactile stimuli, we designed a claw-like mold with 50 × 50 pixels as a detected object (Fig. 5b). The claw-like object is simplified into four stimulus zones including the peripheral flat zone, hair zone, toe pads, and palm pad, which will provide the tactile stimulus inputs of 0, 0.8, 1.6, and 2 kPa to the sensor array. In addition, each stimulus zone contains a few random noise points potentially caused by the attachments that are adhered to the claw of the cat, such as sands. For simplification, these noise points are supposed to be one of three tactile stimuli of 0.8, 1.6, and 2 kPa.

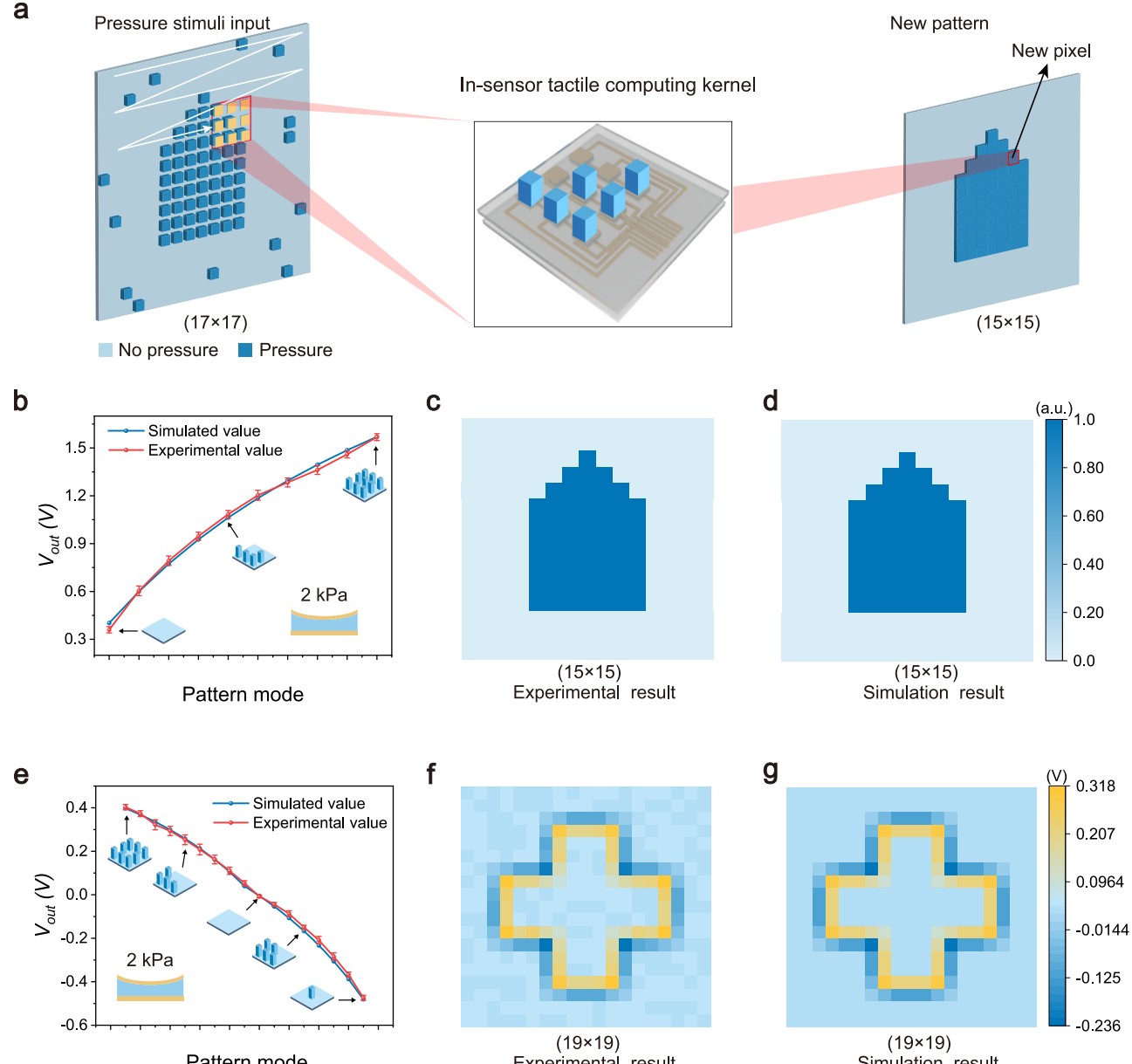

**Fig. 4 | In-sensor tactile computing for binary noise reduction and edge detection. a** Schematic of the capacitive in-sensor tactile computing kernel for processing a detected object with binary stimuli. Local information (3 × 3 pixels) of the object is sensed by the array and simultaneously processed to generate a new pixel. A raster scanning operation with a stride of one needs to obtain a complete information about the object. **b** Experimental $V_{out}$ was consistent with simulated values when the in-sensor tactile computing kernel as an average filter. **c**, **d** Noise reduction results after binarization from experiment (**c**) and simulation (**d**) show good consistency. **e** Experimental $V_{out}$ was consistent with simulated values when the in-sensor tactile computing kernel as the Laplace filter. **f**, **g** Edge detection results from experiment (**f**) and simulation (**g**) are nearly identical.

To execute the noise reduction, we configured the voltage biases of the kernel into an identical value of 3.3 V as an average filter. Figure 5c illustrates the calculated results of the in-sensor tactile averaging computing kernel for all tactile stimulus patterns when the pressure stimuli of 0.8, 1.6, and 2 kPa are applied onto the sensor array, respectively (Supplementary Figs. 10, 14, and 15). For example, the calculated $V_{out}$ values of pattern mode 1 (only one pixel subjected to the pressure stimulus) are 0.46, 0.495, and 0.61 V, respectively, for the pressure stimuli of 0.8, 1.6 or 2 kPa. In other words, a large tactile stimulus will cause a high $V_{out}$ value for the same pattern mode, which is explained by the more charges accumulated on the pressure sensor pixel during the charging phase under a large stimulus. At each kind of pressure stimuli, the calculated $V_{out}$ values of the in-

sensor tactile averaging computing kernel for all tactile stimulus patterns could match the simulated results well (Fig. 4b and Supplementary Fig. 16).

Furthermore, we evaluated the in-sensor tactile computing system for processing the analog tactile inputs with mixed stimulus intensities of 0, 0.8, 1.6, and 2 kPa. Figure 5d shows the calculated results of the computing kernel for several tactile stimulus patterns with the mixture of 1.6 and 2 kPa. As expected, the position change of the sensor pixels subjected to pressure stimuli induces a negligible change in calculated $V_{out}$ value. Figure 5e shows the calculated results for several much more complex tactile stimulus patterns with the mixture of 0.8, 1.6, and 2 kPa. The calculated $V_{out}$ values are also consistent with the simulated results with a maximum deviation of 0.021 V

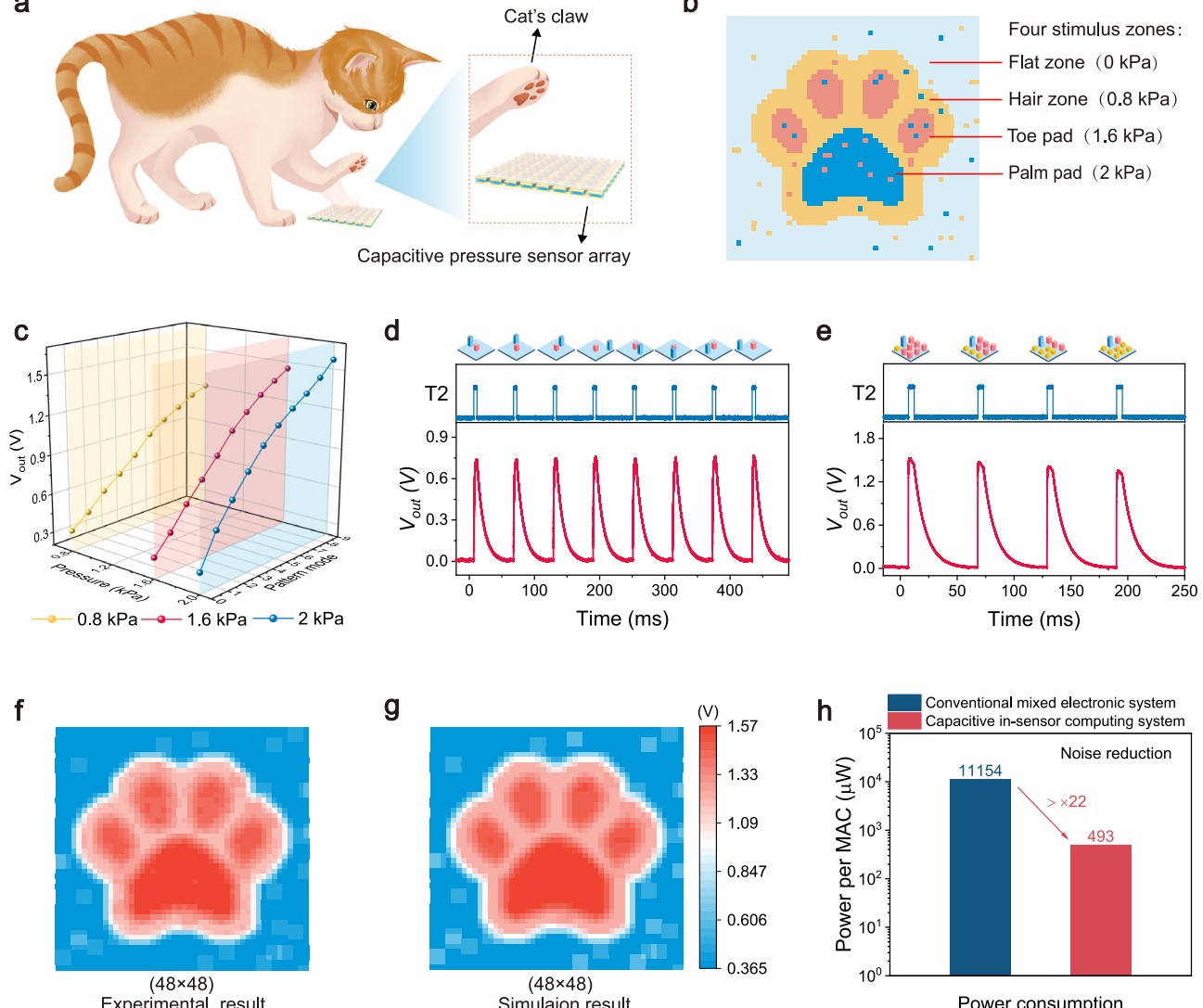

**Fig. 5 | In-sensor tactile computing for analog noise reduction. a** Analog tactile stimuli when a cat's claw contacts the flexible capacitive pressure sensor array. **b** Claw-like noisy mold with multiple tactile stimulus intensities. The mold is simplified to four stimulus zones including flat zone (0 kPa), hair zone (0.8 kPa), toe pad (1.6 kPa), and palm pad (2 kPa). **c** Experimental $V_{out}$ for all tactile stimulus patterns when the pressure stimuli of 0.8, 1.6, and 2 kPa are applied onto the array, respectively. **d, e** Experimental $V_{out}$ for tactile stimulus patterns with the mixture of 1.6 and 2 kPa (**d**) and the mixture of 0.8, 1.6, and 2 kPa (**e**). Yellow pillar: 0.8 kPa; red pillar: 1.6 kPa; blue pillar: 2 kPa; none: 0 kPa. **f, g** Noise reduction results from experiment (**f**) and simulation (**g**). **h** Comparison of power consumption between the capacitive in-sensor tactile computing system and a conventional mixed electronic system.

(Supplementary Note 2). These results demonstrate the capacitive in-sensor tactile computing is feasible for sensing and computing the analog tactile inputs with multiple stimulus intensities.

Similarly, we used the raster scanning operation with a stride of one to process the noise claw-like object (50 × 50 pixels). After being processed, a new claw-like pattern with the reduced noise points (48 × 48 pixels) is obtained (Fig. 5f). The calculated results using the in-sensor tactile computing kernel is consistent with the simulation results (Fig. 5g), showing that the capacitive in-sensor tactile computing system paves a promising pathway for the noise reduction of irregular objects in real applications.

We further evaluated the total power consumption of the capacitive in-sensor tactile computing system, including the power consumed by the capacitive sensors and fixed capacitor $C_0$, and the power consumed by peripheral circuitries such as electrical switches and logic module for switching circuitries. For the noise reduction and edge detection tasks, our system consumed a maximum power of 493 μW and 492 μW for each in-sensor tactile MAC operation

(Fig. 5h, "Methods" and Supplementary Note 4). As comparison, we constructed a conventional mixed electronic system to execute the same task from sensing to computing (Supplementary Fig. 17 and Supplementary Note 4). The conventional mixed electronic system consists of nine capacitive sensors, nine fixed capacitors and nine analogue-to-digital converters (ADC) and a digital MAC module. The ADC and MAC modules were developed as behavioral models using Verilog HDL, we used Vivado software to evaluate the power consumption of the conventional mixed electronic system. The evaluation results indicate that the conventional mixed electronic system has a power consumption of about 11154 μW (Supplementary Note 4). As a result, the power consumption of our system is over 22 times lower than that of the conventional mixed electronic system. Such ultra-low power consumption system is obtained by eliminating the primary energy-consuming components such as ADC and MAC modules between sensing and computing units, making it suitable for power-constrained e-skin applications such as robotics and human-machine interfaces.

## Discussion

In summary, we have demonstrated an in-sensor tactile computing system based on a flexible capacitive pressure sensor array. The system utilizes multiple connected pressure sensor networks to implement the tactile MAC operation based on two physical charging and sharing processes, achieving the in-sensor tactile computing capability. We experimentally fabricated a highly sensitive and highly reliable capacitive pressure sensor array to illustrate the in-sensor tactile computing system for low-level tactile sensory processing tasks, including noise reduction and edge detection. Our system can achieve a maximum power consumption of 493 µW and 492 µW for the noise reduction and edge detection tasks, exhibiting over 22 times lower than a conventional mixed electronic system. Such energy-efficient capacitive in-sensor tactile computing system provides a promising pathway to reshape edge computing for future intelligent applications.

## Methods

### Fabrication of the ionic elastomeric PVA/H$_3$PO$_4$ film

Polyvinyl alcohol (PVA, Mw ~145,000) and phosphoric acid (H$_3$PO$_4$, AR, ≥85%) were purchased from Sigma-Aldrich and Sinopharm Chemical Reagent, respectively. A commercial sandpaper with the roughness of no. 100# was selected as the template. First, the mixture of PDMS precursors (Sylgard 184, Dow Corning) with a weight ratio of 10:1 (the silicone prepolymer: the crosslinker) was poured onto the commercial sandpaper and cured at 60 °C for 4 h. The PDMS film was then peeled off from the sandpaper substrate, acting as a new mold with the microstructured surface. Second, 4 g PVA was dissolved into 36 g deionized water, followed by stirring at 95 °C for 2 h until the PVA dissolved completely. Third, 3 mL H$_3$PO$_4$ was then added into the PVA solution at room temperature and stirred for 2 h. The resultant PVA/H$_3$PO$_4$ solution was then poured onto the PDMS mold and cured at room temperature for 72 h. Finally, a PVA/H$_3$PO$_4$ film with a thickness of ~800 µm was peeled off and then cut into square pieces with the length of 5 mm as the sensing layer of the capacitive pressure sensor array.

### Fabrication of the capacitive pressure sensor array

The WPU stretchable film was chose as the top and bottom substrates due to its high adhesion to Au metal materials, which can improve the reliability of capacitive pressure sensor array. The WPU stretchable film was fabricated from the mixture of WPU emulsion with deionized water and N, N-Dimethylformamide at a weight ratio of 1:1:2. The mixture was dried at 50 °C for 72 h to remove solvent, yielding the WPU elastic substrate[38]. A 50 nm thickness Au layer was directly deposited on the WPU substrate using the shadow mask process via thermal evaporation, acting as the electrodes of the capacitive pressure sensor array. The patterned Au bottom electrode, microstructured PVA/H$_3$PO$_4$ sensing film, and patterned Au top electrode were vertically stacked to form a capacitive pressure sensor array.

### Characterization and measurements

The capacitive sensing characteristics of the pressure sensors were tested using an LCR meter measurement instrument (Changzhou Tonghui Electronic TH2832). The capacitance value was read out at a fixed frequency of 2 kHz. Mechanical measurement equipment (MTS Criterion model C42) and standard weights (1, 2, 5, 10 g) were used to generate external pressure stimuli. An oscilloscope (Tektronix MSO64) was used to monitor the calculated output voltage of the in-sensor tactile computing system.

### Evaluation of power consumption

For our in-sensor tactile computing system, the total power consumption of the system can be categorized into two main sources: the power consumed by the capacitive sensors and fixed capacitor $C_O$, and the power consumed by peripheral circuitries such as electrical switches and logic module for switching circuitries.

The power consumed by the capacitive sensors and the fixed capacitor $C_O$ is calculated as follows: During each MAC operation, the nine capacitive sensors and the fixed capacitor $C_O$ have an effective operating time of only 12 ms, as they are active during the $\Phi_2$ and $\Phi_3$ clock phases (see Supplementary Note 1). For noise reduction and edge detection tasks, the maximum power consumption occurs when all nine capacitive pressure sensors reach their maximum capacitance value (0.86 nF at 2 kPa). In noise reduction task, the maximum power consumption was roughly calculated as:

$$\frac{0.86\,nF \times 9 \times \left((3.3\,V)^2 - (1.57\,V)^2\right) + 10\,nF \times (1.57\,V)^2}{2 \times 12\,ms} = 3.74\,\mu W$$

In the edge detection task, the maximum power consumption was roughly calculated as:

$$\frac{0.86\,nF \times ((-8\,V)^2 + 8 \times (1\,V)^2 - 9 \times (0.04\,V)^2) + 10\,nF \times (0.04\,V)^2}{2 \times 12\,ms} = 2.58\,\mu W$$

The power consumed by electrical switches and logic module for switching circuitries are 70.4 µW and 418.7 µW, respectively (see Supplementary Note 4). Finally, the total power consumption of the system is approximatively 493 µW for noise reduction task and 492 µW for edge detection task.

For a conventional mixed electronic system with the same functionality, the system consists of nine parallel branches and a multiply-accumulate (MAC) module (see Supplementary Fig. 17). Each branch includes a capacitive pressure sensor, a fixed capacitor and an 8-bit ADC. Based on Vivado software platform, we used Verilog HDL to develop the behavioral models of ADC and MAC modules and evaluated the power consumption of the conventional mixed electronic system (Supplementary Note 4). The power consumption of the conventional mixed electronic system for noise reduction and edge detection tasks was estimated as 11154 µW.

## Data availability

The source data generated in this study are provided in the Supplementary Information/Source Data file. Source data are provided with this paper.

## Code availability

The codes used in this study are provided in the Supplementary Information (Supplementary Note 4).

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

## Acknowledgements

This work was in part supported by the National Key Research and Development Program of China: 2021YFB3601200 (M.W.), the National Natural Science Foundation of China: 62488101 (M.L.), U2341218 (M.W.), and 62204052 (J.C.), and we appreciate the support by State Key Laboratory of Integrated Chips and Systems: SKLICS-Z202315 (M.W.).

## Author contributions

M.W. and Y.C. conceived and designed the study. Y.C. designed, fabricated, and characterized the capacitive sensor array devices. J.C. contributed to preparation of the sensor device. Y.C., J.Q., J.Y., and Z.H. designed the readout circuit for data collection. D.Y. and M.Y.L. recorded the optical images. M.Z. performed SEM characterization. C.L. and Y.C. conducted the power consumption estimation. Y.C. and M.W. wrote the paper. Z.W., X.Z., X.C., and E.S. revised the original draft. M.W., Q.L., and M.L. supervised the whole work.

## Competing interests

The authors declare no competing interests.
