## [Transparent Peer Review file · Nature Communications]

Capacitive in-sensor tactile computing

Corresponding Author: Professor Ming Wang

Version 0:

Reviewer comments:

Reviewer #1

(Remarks to the Author)

This manuscript reported a novel capacitive in-sensor tactile computing system based on a capacitive sensor array. The system could detect tactile stimuli in real time and simultaneously process these detected tactile signals with low power consumption. This study is inherently interesting and well-structured. I recommend its publication in Nature Communications with minor revisions. Here are some suggestions for the authors to refine the manuscript.

1. The capacitive sensor array uses the patterned gold (Au) layer as interconnection wires. Will the Au layer under the soft substrate deteriorate under external tactile stimuli? If the Au layer breaks, does it affect the calculation result of the in-sensor tactile computing system?

2. In Figures 3b and 3c, the in-sensor tactile computing system includes the capacitive sensor and the T1, T2, and T3 switches. However, in Figure 3d, the dynamic behaviors of T1 switches and T3 switches were not plotted. Please provide a full behavior timing for all T1, T2, and T3 switches.

3. From Fig 3c and 4d, there seems to be an interval time between the accumulation operation at the $\Phi 3$ phase and the refresh operation at the next $\Phi 1$ phase. What is the role of the interval time? Can the interval time be short to achieve fast processing? How to gradually input large-area pressure images into a 3×3 sensor array?

4. Fig 4b and 4e show the comparison of software and experimental values for different pattern modes. The main text should give more details on the calculated processes and results using software for each pattern mode. Furthermore, the authors should explain the reason for the deviation between software and experimental values.

5. According to Supplementary note 1, a typographical error is in Figure 3d. The time unit should be milliseconds. Should check the manuscript.

6. There are some other studies reported on in-sensor tactile computing system. What are the advantages and disadvantages of this device over other devices? The necessary description need to be provided to further highlight the innovation of this work so that the readers would know this work better.

Reviewer #2

(Remarks to the Author)

In this work, the authors propose a capacitive in-sensor tactile computing system based on a pressure sensor array. By leveraging interconnected sensor networks to perform in-situ analog MAC operations, the array enables both tactile sensing and computing functionalities. Compared with conventional electronic systems, the proposed approach achieves an average power consumption of $3.74 \mu\text{W}$ and $2.58 \mu\text{W}$ for noise reduction and edge detection tasks, respectively—four orders of magnitude lower than typical solutions.

Overall, the novelty of this work is commendable, and the in-sensor tactile computing array demonstrates significant potential for human-machine interface applications. I have a few recommendations that could make the manuscript more clear.

1. In the Introduction, please elaborate on the advantages of using capacitive devices for in-sensor computing. Since pressure sensors can also be realized through piezoresistive approaches, how does capacitive technology compare in terms of integration, sensitivity, power consumption, and overall system complexity?

2. Please compare your proposed tactile sensing-computing approach to other works in the field, especially those involving multimodal sensing systems (e.g., Nat. Commun. 13, 3973 (2022); Adv. Mater. 2200481 (2022); Nat. Commun. 15, 7275 (2024); etc.). Such a comparison will help highlight the advances in device design, power consumption, and array integration presented in your work.

3. The sensor's C2C appears to increase with repeated cycles. What is the underlying mechanism for this progressive increase in response?
4. More data should be provided regarding the D2D variation. This is important for evaluating the feasibility of large-scale applications.
5. The sensor response is reported as a change in capacitance. Are the initial capacitance values of all devices uniform? If not, how might this variation affect overall performance?
6. How does the sensor's performance change under stretching conditions? Please elaborate on any performance degradation or shifts in sensitivity.
7. For the capacitive pressure sensor array, what is the latency for a single MAC operation?
8. Does this capacitive in-sensor computing system offer advantages for large-scale integration? Since your array structure requires multiple switches, what challenges or considerations arise when integrating such a system on a large scale?
9. Although proposed pressure-sensing computation array can perform MAC operations, how is the training process for neural networks implemented on such hardware? Specifically, how are the weights quantized and updated in the physical system, and how does your design address these hardware constraints?

Reviewer #3

(Remarks to the Author)
Please see detailed attachment.

Version 1:

Reviewer comments:

Reviewer #1

(Remarks to the Author)
The authors have thought about the suggestions and comments. There is a suggestion.

The authors listed similar research work only in the Supplementary Information. This makes it easy for readers to overlook relevant research. It is suggested that the authors should introduce these studies in the main text to help readers better understand the relevant progress.

Reviewer #2

(Remarks to the Author)
After a thorough review of the revised manuscript and its supplementary materials, I am satisfied that all of my questions and concerns have been fully addressed. Given the work's significant potential to advance integrated tactile sensing-computing systems, I strongly recommend this manuscript for publication in Nature Communications.

Reviewer #3

(Remarks to the Author)
The authors have addressed all of my comments and I do not have any other concerns. The manuscript is now greatly improved and can be accepted for publication.

Minor

1. Page 16 line 314: The authors did not provide Supplementary Table 5. I suggest the authors to check again the manuscript for similar errors.

Optional

2. The authors envisioned the use of the proposed system to implement e-skin, which would benefit from high-density sensors for a more fine-grained sensing-processing in advanced applications. For the benefit of readers interested in this area, it would be good to add the brief discussion on comment #4 (possibility and feasibility of scaling down for a larger scale sensory integration) in the Supplementary Information.

Detailed responses to reviewers' comments

Reviewer #1:

Comments: *This manuscript reported a novel capacitive in-sensor tactile computing system based on a capacitive sensor array. The system could detect tactile stimuli in real time and simultaneously process these detected tactile signals with low power consumption. This study is inherently interesting and well-structured. I recommend its publication in Nature Communications with minor revisions. Here are some suggestions for the authors to refine the manuscript.*

Response: We sincerely appreciate your highly positive comments on our work. We have provided a one-by-one response to your concerns below.

Q1: *The capacitive sensor array uses the patterned gold (Au) layer as interconnection wires. Will the Au layer under the soft substrate deteriorate under external tactile stimuli? If the Au layer breaks, does it affect the calculation result of the in-sensor tactile computing system?*

Response: Thank you for your comment. In our previous study, we demonstrated that the Au layer on the waterborne polyurethane (WPU) substrate exhibits reliable electrical stability under external tactile stimuli, owing to the strong interfacial adhesion between Au and WPU (*Nat. Commun.* 2024, 15, 1116). As a result, the Au layer on the WPU substrate can serve as an effective stretchable interconnection wire via the microcrack conductive mechanism, demonstrating only a minor increase in electrical resistance under external strain. For example, the electrical resistance of Au stretchable electrode only increases slightly from 42 Ω (0% strain) to 64 Ω (10 % strain), as shown in Figure R1a. Upon releasing the strain, the electrical resistance of Au stretchable electrode will return to its initial value. Hence, the external tactile stimuli will not deteriorate the Au stretchable electrode's performance.

In addition, the slight resistance variation of the Au stretchable electrode does not impair the pressure-capacitance response of the capacitive sensor (Figure R1b). Consequently, the minimal changes in the resistance will not affect the calculation result

of the in-sensor tactile computing system.

Figure R1. **a**, Strain-resistance test of Au stretchable electrode under 10% cyclic tensile deformation. **b**, Pressure-capacitance response of sensors with the Au stretchable electrodes of 42 Ω and 64 Ω , respectively.

Q2: In Figures 3b and 3c, the in-sensor tactile computing system includes the capacitive sensor and the T1, T2, and T3 switches. However, in Figure 3d, the dynamic behaviors of T1 switches and T3 switches were not plotted. Please provide a full behavior timing for all T1, T2, and T3 switches.

Response: According to your suggestion, we have added the complete timing behavior for T1, T2, and T3 switches in Figure 3d (Figure R2). This timing diagram provides a clearer illustration of the operational sequence in our capacitive in-sensor computing system. We have revised the **Figure 3d** and added detailed descriptions of the switching behavior for T1, T2, and T3 switches in the revised **Supplementary Note 1**, as follows: “The turn-on time for all T1, T2 and T3 switches is set to 5 ms. The switching interval between T1, T2 and T3 is set to 2 ms. The interval time between the Φ_3 phase and the next Φ_1 phase is set to 42 ms.”

The corresponding descriptions have been added into the in the revised manuscript. (Lines 203-204, Page 11)

Figure R2. Calculated result of the capacitive in-sensor computing kernel under different tactile stimulus patterns. During each MAC operation, the T3, T1, and T2 switches are sequentially and independently turned on and off.

Q3: From Fig 3c and 3d, there seems to be an interval time between the accumulation operation at the Φ_3 phase and the refresh operation at the next Φ_1 phase. What is the role of the interval time? Can the interval time be short to achieve fast processing? How to gradually input large-area pressure images into a $3 * 3$ sensor array?

Response: Thank you for your valuable comments. The interval time between the Φ_3 phase and the next Φ_1 phase serves to discharge the charges stored on C_0 during Φ_3 phase. While our original manuscript specified a 42 ms interval time (Figure 3c and revised Figure 3d) for gradual C_0 discharge, this duration can actually be reduced to 1 ms or less. This is because the T3 switch will turn on during the next Φ_1 phase to completely discharge C_0 , enabling faster processing. As shown in Figure R3, the voltage across C_0 rapidly drops to zero as the T3 switch turns on in the next Φ_1 phase.

In this work, we employed a raster scanning (sliding) approach to input large-area pressure stimuli into the 3×3 sensor array. For each operation, a 3×3 pixels segment of the large-area pressure stimuli is captured and processed by the in-sensor computing array. After the operation, we slid the array with a stride of one and an adjacent

segment (3×3 pixels) is input until all pressure stimuli are processed, as shown in Figure 4a.

To make it clear, we have revised **Figure 3d** and added Figure R3 has been added as **Supplementary Figure 7a** in the revised Supplementary information. The corresponding discussion has been added in the revised manuscript. (*Lines 203-204, Page 11*)

Figure R3. Timing diagram of the capacitive in-sensor computing kernel to implement the MAC operation. Setting the interval time to 1 ms, the voltage across C_0 rapidly drops to zero as the T3 switch turns on in the next Φ_1 phase.

Q4: Fig 4b and 4e show the comparison of software and experimental values for different pattern modes. The main text should give more details on the calculated processes and results using software for each pattern mode. Furthermore, the authors should explain the reason for the deviation between software and experimental values.

Response: Thank you for your comment. Fig 4b and 4e show the comparison of simulated and experimental values for different pattern modes. We have updated all text description in the revised version. For tactile stimulus pattern modes, the simulated results can be calculated using Equation 4, which have taken the leakage current and parasitic capacitance effects into account (Supplementary Note 2).

$$V_{out} = \frac{Q_{sum}}{\sum_{i=1}^9 C_i + C_0} + V_{base} = \frac{\sum_{i=1}^9 C_i \times V_i}{\sum_{i=1}^9 C_i + C_0} + V_{base}, \quad i=1, 2 \dots 9 \quad (4)$$

The baseline voltage is 0.13 V for noise reduction task and 0.04 V for edge detection task, respectively.

Calculated processes for noise reduction task (Fig 4b):

For pattern mode 0: With all nine sensors at 0 kPa pressure, the simulation calculated value is:

$$V_{out} = \frac{\sum_{i=1}^9 C_i \times V_i}{\sum_{i=1}^9 C_i + C_0} + 0.13 V = \frac{(0 \times 0.86 \text{ nF} + 9 \times 0.1 \text{ nF}) \times 3.3 V}{0 \times 0.86 \text{ nF} + 9 \times 0.1 \text{ nF} + 10 \text{ nF}} + 0.13 V$$

$$= 0.402 V$$

For pattern mode 1: With one sensor at 2 kPa and other eight sensors at 0 kPa, the simulation calculated value is:

$$V_{out} = \frac{\sum_{i=1}^9 C_i \times V_i}{\sum_{i=1}^9 C_i + C_0} + 0.13 V = \frac{(1 \times 0.86 \text{ nF} + 8 \times 0.1 \text{ nF}) \times 3.3 V}{1 \times 0.86 \text{ nF} + 8 \times 0.1 \text{ nF} + 10 \text{ nF}} + 0.13 V$$

$$= 0.600 V$$

Other patterns follow the same calculation method. All simulation calculated values are listed in the **Table R1**.

Table R1 Simulation calculated values with the in-sensor tactile computing kernel as the average filter under the pressure of 2 kPa.

Pattern mode	V_{out} (V)
0	0.402
1	0.600
2	0.773
3	0.926
4	1.063
5	1.185
6	1.295
7	1.395
8	1.487
9	1.57

Calculated processes for edge detection task (Fig 4e):

For pattern mode A0: With eight sensors at 2 kPa and central sensor at 0 kPa, the simulation calculated value is:

$$V_{out} = \frac{\sum_{i=1}^9 C_i \times V_i}{\sum_{i=1}^9 C_i + C_0} + 0.04 V = \frac{(8 \times 0.86 \text{ nF}) \times 1 V + 1 \times 0.1 \text{ nF} \times (-8) V}{8 \times 0.86 \text{ nF} + 1 \times 0.1 \text{ nF} + 10 \text{ nF}} + 0.04 V$$

$$= 0.398 V$$

For pattern mode A1: With seven sensors at 2 kPa, one sensor at 0 kPa and central sensor at 0 kPa, the simulation calculated value is:

$$V_{out} = \frac{\sum_{i=1}^9 C_i \times V_i}{\sum_{i=1}^9 C_i + C_0} + 0.04 V = \frac{(7 \times 0.86 \text{ nF} + 0.1 \text{ nF}) \times 1 V + 1 \times 0.1 \text{ nF} \times (-8) V}{7 \times 0.86 \text{ nF} + 2 \times 0.1 \text{ nF} + 10 \text{ nF}} + 0.04 V$$

$$= 0.368 V$$

Other patterns follow the same calculation method. All simulation calculated values are listed in the **Table R2**.

Table R2 Simulation calculated values when the in-sensor tactile computing kernel as the Laplace filter under the pressure of 2 kPa.

Pattern mode	V_{out} (V)
A0	0.398
A1	0.368
A2	0.335
A3	0.299
A4	0.258
A5	0.213
A6	0.162
A7	0.105
A8	0.04
A9	-0.005
A10	-0.054
A11	-0.107
A12	-0.167

A13	-0.233
A14	-0.306
A15	-0.388
A16	-0.481

As shown in Figs. 4b and 4e, the simulated and experimental values show strong agreement after accounting for leakage current and parasitic capacitance effects (see Supplementary Note 2). The minor remaining deviations likely stem from sensor variations, because the simulated calculation assumes that all capacitive pressure sensors have uniform sensing characteristics.

Tables R1 and R2 have been included as Supplementary Table 2 and Supplementary Table 3, respectively, in the revised Supplementary Information. Additionally, we have restructured Supplementary Note 2 to detail the simulation methodology.

Q5: *According to Supplementary note 1, a typographical error is in Figure 3d. The time unit should be milliseconds. Should check the manuscript.*

Response: We have updated the unit labels in Figure 3d to milliseconds. We appreciate your careful review.

Q6: *There are some other studies reported on in-sensor tactile computing system. What are the advantages and disadvantages of this device over other devices? The necessary description needs to be provided to further highlight the innovation of this work so that the readers would know this work better.*

Response: Thank you for your insightful suggestion. To better highlight the innovation of our study, we have provided a detailed comparison between our work and other tactile sensing-computing systems, including in-sensor and near-sensor tactile computing systems, as summarized in **Table R3**.

From this table, it is evident that the key distinction between our work and existing

in-sensor tactile computing systems lies in the implementation of computing functionality and its reconfigurability. For example, previous in-sensor tactile computing systems primarily rely on novel sensing structures to fuse tactile signals and spatial information into fusion signals, which are then decoded via software. Similarly, prior multimodal sensing-computing systems typically employ an integrated device structure, where temperature signals (pressure signals) are perceived by a Mott memristor (pressure sensor) and encoded into spike trains using the Mott memristor. These encoded spike trains are subsequently decoded by software. In these systems, the core computing functionality is limited to signal conversion, implemented by physical devices. Moreover, the computing capability cannot be reconfigured.

Table R3 Comparison between our work and other in-sensor and near-sensor tactile computing systems.

Type	Device design		Computing architecture	Computing functionality	Functional reconfiguration	Reference
	Sensing unit	Computing unit				
Multimode-fused spiking neuron	Pressure sensor (Pressure), Memristor (Temperature)	Mott Memristor	Near/In-sensor computing	Fusing signals (spike encoding)	No	Adv. Mater. 2022, 34, 2200481
Crossmodal sensory neuron	Pressure sensor (Pressure), Memristor (Temperature)	Mott Memristor	Near/In-sensor computing	Fusing signals (spike encoding)	No	Nat. Commun. 2024, 15, 7275
Calibratable sensory neuron	Pressure sensor (Pressure), Light sensor (Light), Memristor (Temperature)	Mott Memristor	Near-sensor computing	Fusing signals (spike encoding)	No	Nat. Commun. 2022, 13, 3973
Artificial tactile near-sensor computing unit	Triboelectric sensor	Synaptic transistor	Near-sensor computing	Threshold detection	No	Adv. Funct. Mater. 2024, 2401913
Tactile near-sensor analogue computing	Pressure sensor	Nonvolatile memristor	Near-sensor computing	Vector-matrix multiplication	Yes (Averaging filter, Laplacian filter)	Adv. Mater. 2022, 34, 2201962
IR touch sensor	Pressure sensor		In-sensor computing	Fusing signals	No	Adv. Funct. Mater. 2024, 34, 2411331
LPI tactile sensor	Pressure sensor		In-sensor computing	Fusing signals	No	Adv. Mater. 2024, 2407329
Capacitive in-sensor tactile computing	Pressure sensor		In-sensor computing	Vector-matrix multiplication	Yes (Averaging filter, Laplacian filter)	This work

In contrast, our capacitive in-sensor tactile computing system enables physical vector-matrix multiplication (VMM) operations through the configuration of the sensor array. The computing operation is fundamentally different from the computing functionalities reported in previous works, as it represents a critical step in artificial

neural network (ANN) algorithms (*Nature* 2020, 579, 62-66; *Nat. Mater.* 2023,22, 1499-1506). Moreover, our system allows the VMM operation to be reconfigured into different kernels or filters, enabling its application to diverse tasks, such as noise reduction, edge extraction and sharpness. This reconfigurable computing functionality demonstrates its advantages over other reported in-sensor and near-sensor tactile computing systems.

We have supplemented the revised manuscript with detailed textual descriptions to highlight the advantages of our work (*Lines 59, Page 4*). Additionally, **Table R3** has been added as **Supplementary Table 1** in the revised supplementary information.

Reviewer #2:

Comments: *In this work, the authors propose a capacitive in-sensor tactile computing system based on a pressure sensor array. By leveraging interconnected sensor networks to perform in-situ analog MAC operations, the array enables both tactile sensing and computing functionalities. Compared with conventional electronic systems, the proposed approach achieves an average power consumption of 3.74 μW and 2.58 μW for noise reduction and edge detection tasks, respectively—four orders of magnitude lower than typical solutions. Overall, the novelty of this work is commendable, and the in-sensor tactile computing array demonstrates significant potential for human-machine interface applications. I have a few recommendations that could make the manuscript more clear.*

Response: We sincerely appreciate your positive feedback on our work. According to your comments, more comprehensive experimental studies and detailed analyses have been conducted. Our responses to your specific comments one by one are shown as follows.

Q1: *In the Introduction, please elaborate on the advantages of using capacitive devices for in-sensor computing. Since pressure sensors can also be realized through piezoresistive approaches, how does capacitive technology compare in terms of integration, sensitivity, power consumption, and overall system complexity?*

Response: Thank you for your comment. We have added a relevant description on the advantages of capacitive sensors in the Introduction section, as follows: “Among various tactile sensing technologies, capacitive-type sensors offer the advantages of high sensitivity, excellent stability, fast response time and lower power consumption.” Capacitive and piezoresistive sensors are both widely studied pressure sensor technologies due to their simple structures and excellent performance. Significant research efforts have focused on enhancing their sensing capabilities and integration potential for practical applications. We selected capacitive sensors for in-sensor computing primarily because of their inherent suitability for implementing vector-matrix multiplication (VMM) through a novel capacitive network architecture. The

network architecture is different from the traditional capacitive sensing array. In our network architecture, the top electrode of each sensor pixel is connected to two individual electrical switches (T1 and T2), and the bottom electrodes of all sensor pixels are grounded. Multiple pressure sensor pixels in the array are interconnected via their corresponding T2 switches, forming a capacitive in-sensor tactile computing subregion. Based on this architecture, tactile vector-matrix multiplication (VMM) operations can be performed in the charge domain based on two physical charging and sharing processes. However, implementing similar functionality with piezoresistive sensors would necessitate a completely different network architecture that has yet to be developed, making system complexity and integration feasibility dependent on future piezoresistive in-sensor computing designs.

We have restructured the Introduction section to improve clarity and added a relevant description on the advantages of capacitive sensors. The corresponding descriptions have been added into the in the revised manuscript. (*Lines 61-63, Page 4*)

Q2: *Please compare your proposed tactile sensing-computing approach to other works in the field, especially those involving multimodal sensing systems (e.g., Nat. Commun. 13, 3973 (2022); Adv. Mater. 2200481 (2022); Nat. Commun. 15, 7275 (2025); etc.). Such a comparison will help highlight the advances in device design, power consumption, and array integration presented in your work.*

Response: Thank you for your constructive suggestion. In accordance with your suggestions, we have provided a detailed comparison between our work and other tactile sensing-computing systems, including these mentioned multimodal sensing systems, as summarized in **Table R3**.

Unlike most existing tactile sensing-computing systems, our approach adopts an in-sensor computing architecture, where sensing and computing units employ the same physical devices. In contrast, conventional and near-sensor architectures rely on separated sensing and computing units for these functions. Hence, our approach can minimize the transfer of redundant data between sensing units and computing units, reducing the complexity of data processing and conversion.

Compared to the state-of-the-art in-sensor tactile systems including multimodal ones, our system still shows the key advantage in the implementation of computing functionality and its reconfigurability, as the response to Q6 from reviewer 1. In these multimodal sensing systems, pressure signals (temperature signals) are perceived by the pressure sensor (Mott memristor) and encoded into spike trains using a Mott memristor. These encoded spike trains are subsequently decoded by software. Similarly, the recently reported in-sensor tactile computing systems primarily rely on novel sensing structures to fuse tactile signals and spatial information into fusion signals, which are then decoded by software. In these systems, the core computing functionality is limited to signal conversion, without the capability for reconfiguration. In contrast, our system enables physical VMM operations through the design of the sensor array. The computing operation is fundamentally different from the computing functionalities reported in previous works, as it represents a critical step in ANN algorithms. Moreover, our system allows the VMM operation to be reconfigured into different kernels or filters, enabling its application to diverse tasks, such as noise reduction, edge extraction and sharpness. This reconfigurable computing functionality demonstrates its advantages over other reported in-sensor and near-sensor tactile computing systems.

As to power consumption, prior multimodal sensing systems typically report per-spike power (calculated approximately from the $V-t$ and $I-t$ curves), which cannot represent the total energy cost across the whole sensing-to-computing pipeline. In contrast, our work given a system-level power consumption per VMM operation, encompassing the entire sensing-to-computing process. A direct comparison of power consumption between our work and other most reported works is inequivalent, due to the differentiation of sensing-to-computing functionalities. Hence, in our work, we benchmark our system against a conventional mixed electronic system with similar functionality and input signal parameters, demonstrating a 22-fold reduction in power consumption.

In addition, our proposed in-sensor computing array is able to achieve large-scale integration by replacing these discrete switches with transistors, as the response to Q8.

We have added detailed textual descriptions in the revised manuscript to highlight the advantages of our work (*Lines 59, Page 4*). Additionally, **Table R3** has been added as **Supplementary Table 1**.

Table R3 Comparison between our work and other in-sensor and near-sensor tactile computing systems.

Type	Device design		Computing architecture	Computing functionality	Functional reconfiguration	Reference
	Sensing unit	Computing unit				
Multimode-fused spiking neuron	Pressure sensor (Pressure), Memristor (Temperature)	Mott Memristor	Near/In-sensor computing	Fusing signals (spike encoding)	No	Adv. Mater. 2022, 34, 2200481
Crossmodal sensory neuron	Pressure sensor (Pressure), Memristor (Temperature)	Mott Memristor	Near/In-sensor computing	Fusing signals (spike encoding)	No	Nat. Commun. 2024, 15, 7275
Calibratable sensory neuron	Pressure sensor (Pressure), Light sensor (Light), Memristor (Temperature)	Mott Memristor	Near-sensor computing	Fusing signals (spike encoding)	No	Nat. Commun. 2022, 13, 3973
Artificial tactile near-sensor computing unit	Triboelectric sensor	Synaptic transistor	Near-sensor computing	Threshold detection	No	Adv. Funct. Mater. 2024, 2401913
Tactile near-sensor analogue computing	Pressure sensor	Nonvolatile memristor	Near-sensor computing	Vector-matrix multiplication	Yes (Averaging filter, Laplacian filter)	Adv. Mater. 2022, 34, 2201962
IR touch sensor	Pressure sensor		In-sensor computing	Fusing signals	No	Adv. Funct. Mater. 2024, 34, 2411331
LPI tactile sensor	Pressure sensor		In-sensor computing	Fusing signals	No	Adv. Mater. 2024, 2407329
Capacitive in-sensor tactile computing	Pressure sensor		In-sensor computing	Vector-matrix multiplication	Yes (Averaging filter, Laplacian filter)	This work

Q3: *The sensor's C2C appears to increase with repeated cycles. What is the underlying mechanism for this progressive increase in response?*

Response: This progressive increase in response of our capacitive sensor is attributed to the hysteresis of the polyvinyl alcohol/phosphoric acid (PVA/H₃PO₄) sensing film during repeated cycles. Our capacitive sensor has a multi-layer stacked structure consisting of a microstructured PVA/H₃PO₄ sensing film and stretchable Au electrodes. As an elastomeric polymer, the PVA/H₃PO₄ film will inherently exhibit a certain degree of hysteresis during repeated cycling tests, as confirmed by cyclic stress-strain test (Figure R4). The hysteresis comes from finite amount of time needed for the polymer chains to revert back to their original positions, thus resulting in the progressive increase in response (*Adv. Mater. 2019, 31, 1904765; Adv. Funct. Mater. 2024, 34,*

2316346).

Figure R4 Cyclic stress-strain test of elastomer PVA/H₃PO₄ films.

Q4: *More data should be provided regarding the D2D variation. This is important for evaluating the feasibility of large-scale applications.*

Response: Thanks for your suggestion. We have supplemented pressure-capacitance response tests for 30 devices, exhibiting a good device-to-device (D2D) uniformity, as shown in Figure R5a. To quantify the D2D variation, we calculated the variation coefficient ($C_v = \sigma/\mu$, σ is the standard deviation and μ is the mean value) of sensors under the pressure of 0, 0.8, 1.6 and 2 kPa, as shown in Figure R5b. The C_v value of 30 sensors at the pressure of 0, 0.8, 1.6 and 2 kPa was 6.98%, 4.95%, 4.59% and 4.41%, respectively, demonstrating our sensors exhibit low D2D variation.

Figure R5 has been added as **Supplementary Figure 4** in the revised supplementary information, with corresponding text description added. (*Lines 143-144, Page 8*)

Figure R5 a, Capacitance-pressure response of 30 sensor devices. The applied pressure ranges from 0 to 4 kPa. **b**, Variation coefficient of sensor under the pressure of 0, 0.8, 1.6 and 2 kPa.

Q5: *The sensor response is reported as a change in capacitance. Are the initial capacitance values of all devices uniform? If not, how might this variation affect overall performance?*

Response: To evaluate the uniformity of the initial capacitance values of all devices, we extracted the variation coefficient of 30 sensors under the pressure of 0 kPa (from Figure R5a), as shown in Figure R6. The initial capacitance values of the 30 sensors range from approximately 85 to 115 pF, with an average value of 100.68 pF. We use the simulated calculation V_{out} to evaluate the impact of initial capacitance variations on overall performance, based on Equation 4 in the Supplementary Note 2.

For the noise reduction task under extreme conditions, all capacitive sensors are either 85 pF or 115 pF. When all sensors are 85 pF, the simulated calculation V_{out} is 0.365 V for tactile pattern mode 0; when all sensors are 115 pF, the simulated calculation V_{out} is 0.439 V for tactile pattern mode 0. According to Table R1 (Response to Q4 from Reviewer 1), V_{out} ranges from 0.402 V to 1.57 V as the tactile pattern shifts from mode 0 to mode 9. Similar behavior was observed for the edge detection task. Despite the variation between these extreme cases and the calculated values, the overall calculated performance remains acceptable. In the future, achieving a more uniform capacitive sensor array would further improve consistency

in future implementations.

Figure R6 Distribution range of initial capacitance values for 30 sensor devices.

Q6: *How does the sensor's performance change under stretching conditions? Please elaborate on any performance degradation or shifts in sensitivity.*

Response: To address this issue, we conducted pressure-capacitance response tests under stretching conditions. In the test, we stretched two capacitive sensors (#1 and #2) to 5% strains to evaluate their performance changes. Both capacitive sensors exhibited similar pressure-capacitance responses. As shown in Figure R7a, under tensile conditions, the sensitivity of the capacitive sensor is enhanced, increasing from $0.37 \text{ nF}\cdot\text{kPa}^{-1}$ at 0% strain to $0.43 \text{ nF}\cdot\text{kPa}^{-1}$ at 5% strain. The phenomenon can be explained by the fact that the thickness of the sensing film layer decreases as the increase of tensile strains, leading to a higher capacitance value response under the same pressure (*Nat Electron* 2018, 1, 314–321; *ACS Appl. Mater. Interfaces* 2020, 12, 27961–27970). Despite these changes, both sensors maintained reliable performance, demonstrating robustness under deformation.

Figure R7 a, Schematic illustrations of the pressure sensor operating under unstretched (left) and stretched (right) states. **b**, Capacitance-pressure response of the capacitive sensor at 0 and 5% strains.

Q7: For the capacitive pressure sensor array, what is the latency for a single MAC operation?

Response: As shown in Fig 3c and Supplementary Note 1, each MAC operation begins when the T3 switch turn on and concludes when the T2 switches turn off. In our work, the latency time between the 1st and 2nd MAC operation is about 42 ms (as shown in Figure R8). The time latency can be reduced to 1 ms or less for faster processing (Figure R3), as the response to Q3 from Reviewer 1. This is because the T3 switch will turn on during the next $\Phi 1$ phase, which can completely discharge C_0 . As shown in Figure R3, the voltage across C_0 rapidly drops to zero as the T3 switch turns on in the next $\Phi 1$ phase.

To make it clearer, we have updated **Figure 3d** to indicate the latency time. The corresponding discussion of latency reduction has been added to both the revised main text and **Supplementary Figure 7**. (Lines 203-206, Page 11)

Figure R8. Timing diagram of the capacitive in-sensor computing kernel to implement the MAC operation **a**, latency for a single MAC operation is 42 ms.

Q8: *Does this capacitive in-sensor computing system offer advantages for large-scale integration? Since your array structure requires multiple switches, what challenges or considerations arise when integrating such a system on a large scale?*

Response: Thank you for your comment. In our current implementation, the capacitive in-sensor computing system employs multiple switches to regulate the physical charging and sharing processes of capacitive sensors, enabling multiplication and accumulation (MAC) operations. However, these switches can be replaced using transistors exhibiting switching characteristics, providing a feasible approach for large-scale integration.

We have added relevant discussions in the revised manuscript. (Lines 226-227, Page 12).

Q9: *Although proposed pressure-sensing computation array can perform MAC operations, how is the training process for neural networks implemented on such hardware? Specifically, how are the weights quantized and updated in the physical system, and how does your design address these hardware constraints?*

Response: Thank you for your comment. In the current design, our proposed in-

sensor tactile computing system can only be used for inference rather than training. During inference, all the elemental values (V_1, V_2, \dots, V_n) of the voltage vector V are predefined as specific positive or negative values, and the capacitive sensor matrix C are used to perceive and simultaneously process the tactile stimuli. This enables direct hardware implementation of sensory computations like noise reduction, edge extraction, and sharpness enhancement. The weights can be quantized in this physical system by adjusting the values of the voltage vector.

Reviewer #3:

Comments: *This article by Chen et al. reports on a tactile in-sensor computing system using an array of capacitive sensors. With proper periphery circuitries, the sensor array also functions as a switched-capacitor circuit and can perform analog MAC operation via charge sharing. Convolution-based preprocessing of tactile sensory signals such as noise filtering and edge detection was demonstrated.*

The fabricated sensors seem to have very good uniformity. The idea of using capacitive array to perform MAC operations is not new, but incorporating capacitive sensor units into in-sensor MAC hardware is interesting. However, one major issue I have with this work is that the operation of the capacitive in-sensor array as demonstrated will face serious challenges when deployed in real-world applications (see comments #1-2), ultimately limiting its use to toy problems only. Several other issues are also listed below.

Response: Thank you for your valuable comments and high praise for the innovation in our work. In the current implementation, the convolution-based preprocessing task for tactile signals requires sliding the sensor array across the tactile input plane. To overcome this issue, we have conceived a feasible strategy that employed a large-area sensor array with properly sequenced electrical switches to process the entire tactile information of the object, eliminating mechanical movement and enhancing practical applicability. Detailed explanations can be found in our responses to comments #1. The processing time can be significantly reduced in the additional experiments, as detailed in our response to comment #2.

Additionally, we have carefully addressed each of your specific comments, as outlined below. These revisions have significantly enhanced the overall quality of the manuscript.

Q1: *Based on the demonstration, the capacitance array acts as a convolution kernel. To process a single 2D tactile input, the array needs to be slid in both x and y directions. This does not seem like an elegant/logical/practical implementation: imagine having to physically slide the array under a cat's paw just to obtain a processed static tactile image.*

Response: Thank you for your very insightful comment. As you mentioned, our current implementation requires sliding the capacitive in-sensor computing array along both x- and y-axes to process a large-scale 2D tactile input. However, this sliding operation can be eliminated by adopting a large-scale capacitive sensor array with the same architecture. In other words, such an array can preprocess the entire tactile input through multiple MAC operations without sliding operations, making the in-sensor computing array more practical for real-world applications. Detailed explanations are shown as follows.

Taking the preprocessing of a $m \times n$ tactile input as an example, we employ a corresponding $m \times n$ capacitive in-sensor computing array. In the array, the T1 switches of each capacitive sensor unit are connected to corresponding input voltage biases ($V_{11} \dots V_{mn}$), while all T2 switches are interconnected together, ensuring that all capacitive sensing units are in parallel with a fixed capacitor C_0 , as shown in Figure R9. When a real object with $m \times n$ pixels is pressed onto this array, the $m \times n$ capacitive in-sensor computing array simultaneously detects all tactile input, which is then processed in sequential 3×3 -pixel segments via changing the clock sequence. Firstly, a 3×3 segment (magenta dotted region) is processed by controlling the electrical switch groups (T1, T2) of the 9 capacitive sensor units in that segment, outputting a corresponding V_{out} value. The switches of the remaining sensors unit remain in the off state and thus not participating in the computing procedure. Then, by shifting the clock sequence, an adjacent segment (green dotted region) is processed similarly. This sequential approach allows the $m \times n$ tactile input processing without sliding operations, as the entire array captures the stimuli at once, and computation proceeds segment by segment via electrical switches.

We have added the corresponding text descriptions in revised main text to address this concern (*Lines 221-226, Page 11-12*). Figure R9 has been added as **Supplementary Figure 9** in the revised Supplementary information.

Figure R9 Circuit schematic of the $m \times n$ capacitive sensor array for convolutional computation on a $m \times n$ pixels plane, which eliminates the need for the capacitive sensor array to slide across the pixel plane.

Q2: The decay time C_0 is on the order of tens of seconds in Fig. 3d, but on the order of tens of milliseconds in Fig. 5d, e. Did the authors change any circuit parameters, or did they unintentionally mislabel the units? Also, if tens of milliseconds is needed to complete one MAC operation, the time needed to complete the preprocessing of the entire tactile image (see comment #1) would be unacceptably long. How should this issue be addressed?

Response: We sincerely appreciate your thorough review and for catching the labeling error. The unit label for the x-axis in **Figure 3d** should be labeled in milliseconds, and we have corrected this in the revised version.

As to processing time optimization, both the electrical switches' turn-on time and

the interval time between two adjacent MAC operations can be reduced (Figure R10). For example, we have added new experiments to demonstrate that the interval time (42 ms) between two operations could be reduced to 1 ms (Figure R10b). This is because the T3 switch will turn on during the next Φ_1 phase to completely discharge C_0 , enabling faster processing. As demonstrated in Figure R10b, the voltage across C_0 rapidly drops to zero as the T3 switch turns on in the next Φ_1 phase. Meanwhile, the turn-on time for all T1, T2 and T3 switches can be shortened to further improve speed (Figure R10a). Hence, the overall processing time can be further reduced by optimizing the switching circuits.

We have added two new experiments to address this concern. Figure R10 has been included as Supplementary Figure 7, and corresponding text descriptions has added in the revised version. (*Lines 203-206, Page 11*)

Figure R10. a, Timing diagram of the capacitive in-sensor computing kernel to implement the MAC operation. **b**, The electrical switches' turn-on time and the interval time between adjacent MAC operations can be further reduced.

Q3: How should the value of C_0 be determined? Is choosing a value such that $C_0 \gg \sum C_i$ the best strategy, perhaps in terms of SNR, power consumption, etc.? I believe that a smaller C_0 may increase the readout voltage range, which might be more desirable. Besides, with the current C_0 of 10 nF, $C_0 \gg \sum C_i$ does not really hold in most cases. Even the calculations in Supplementary Note 2 do not use this assumption. It would be

less confusing if the authors simply omit this assumption.

Response: Thank you for your comment. In our capacitive in-sensor tactile computing system, the multiplication-and-accumulation (MAC) operation output corresponds to the total charges (Q_{sum}) generated across all pressure sensor pixels during the Φ_2 phase.

$$Q_{sum} = \sum_{i=1}^n Q_i = \sum_{i=1}^n C_i \times V_i, \quad i=1,2\dots n$$

The fixed capacitor C_0 is connected in parallel to the pressure sensor pixels in the array, serving as the readout component of Q_{sum} . As the reviewer mentioned, a smaller C_0 could increase the readout voltage range, but it would be insufficient to readout the real Q_{sum} . If C_0 is too large, it will result in an excessively small V_{out} value, which is unfavorable for reliable sampling, degrading the SNR. Although the assumption $C_0 \gg \sum C_i$ can cause a high readout accuracy on Q_{sum} , we still chose a fixed capacitor of 10 nF as C_0 in our implementation, which is a balanced trade-off. To avoid confusing, we remove this assumption in the revised version. In addition, we reorganized the related descriptions in the main text and Supplementary Note 2, and added the corresponding discussion on the selection of C_0 in the Supplementary Notes 1. (Lines 102-106, Page 6)

Q4: *How small can each capacitive sensor be physically scaled to while maintaining the appropriate absolute capacitance, range of capacitance change, sensitivity, resistance to noise, etc.? How significant is the increase in parasitic capacitances between adjacent cells upon scaling down and how do they affect computations? These are crucial when trying to achieve higher spatial resolution for a much wider range of applications.*

Response: We appreciate the reviewer's comment. To clarify this comment, we fabricated capacitive sensors with dimensions of $3 \times 3 \text{ mm}^2$ and $1 \times 1 \text{ mm}^2$, and compared them with the original $5 \times 5 \text{ mm}^2$ capacitive sensor to examine changes in absolute capacitance and sensitivity as the size of the sensor was physically scaled, as shown in Figure R11. When the physical size of the capacitive sensor is reduced, the absolute

capacitance value and sensitivity will decrease accordingly due to the reduction in area. Therefore, the sensor's noise resistance would decrease with its physical size.

Figure R11 Capacitance-pressure response of the capacitive sensors with different physical sizes.

To evaluate the parasitic capacitance effect between adjacent cells, a coplanar microstrip model is employed (*IEEE Trans. Microw. Theory Tech.*, 59, (2001); *Prog. Electromagn. Res. Lett*, Vol. 54, 79–84, (2015)). Taking a 3×3 capacitive sensor array as an example, the parasitic capacitance of the central sensor cell primarily arises from adjacent and diagonal directions, as shown in Figure R12. Given that the electrode thickness of the capacitive sensors is only 50 nm, the parasitic capacitance of the central sensor cell is predominantly attributed to edge effects induced by fringing electric fields.

Adjacent parasitic capacitance: In the capacitive sensor array, both the size w and spacing s of the sensor cells are 5 mm, and the vacuum dielectric constant ϵ_0 is $8.85 \times 10^{-12} \text{ F}\cdot\text{m}^{-1}$, and the relative permittivity ϵ_r of air is found to be 1. The coupling capacitance caused by a single adjacent sensor cell is:

$$C_{edge} = \frac{4\epsilon_0\epsilon_r \cdot w \cdot K(k)}{K(k')} = \frac{4\epsilon_0\epsilon_r \cdot w \cdot K\left(\frac{w}{w+2s}\right)}{K\left(\sqrt{1-\left(\frac{w}{w+2s}\right)^2}\right)} = \frac{4 \times 8.85 \times 10^{-12} \text{ F}\cdot\text{m}^{-1} \times 1 \times 5 \text{ mm} \times K\left(\frac{5 \text{ mm}}{5 \text{ mm} + 10 \text{ mm}}\right)}{K\left(\sqrt{1-\left(\frac{5 \text{ mm}}{5 \text{ mm} + 10 \text{ mm}}\right)^2}\right)}$$

$$= 4 \times 8.854 \times 10^{-12} \times 1 \times 5 \text{ mm} \times 0.608 \approx 0.108 \text{ pF}$$

The parameters $K(k)$ and $K(k')$ are the first complete elliptic integrals, respectively. Their ratio can be obtained from standard mathematical tables or numerical.

Diagonal parasitic capacitance: The diagonal capacitive sensing cell are located at a distance of $5\sqrt{2} \text{ mm}$ from the central sensor unit. Accordingly, the coupling capacitance contributed by each diagonal cell can be expressed as:

$$C_{diag} \approx \frac{C_{edge}}{\sqrt{2}} = \frac{0.108 \text{ pF}}{\sqrt{2}} \approx 0.076 \text{ pF}$$

Since there are four neighboring sensor cells along the linear directions and another four along the diagonal directions, the total parasitic capacitance of the central sensor cell can be expressed as follows:

$$C_{parasitic} = (C_{edge} + C_{diag}) \times 4 = (0.108 \text{ pF} + 0.076 \text{ pF}) \times 4 = 0.736 \text{ pF}$$

The ratio of the sensor cell's parasitic capacitance to its initial capacitance:

$$\frac{C_{parasitic}}{C_{initial}} = \frac{0.736 \text{ pF}}{100 \text{ pF}} < 1\%$$

According to the calculation approach mentioned above, the parasitic capacitance of the central sensing unit in a capacitive sensor array with both size and spacing of 3 mm and 1 mm is approximately 0.44 pF and 0.087 pF, respectively. As the size of the capacitive sensor cell scaling down, the parasitic capacitance also decreases proportionally, where the parasitic capacitance accounts for less than 1% of the initial value of the capacitance. Therefore, the parasitic capacitance has a negligible effect on the computations.

Figure R12. Positional distribution of the central sensor cell with adjacent and

diagonally oriented sensor cells in a 3×3 capacitive sensor array.

Q5: *Fig. 1c and Supplementary Fig. 1 is misleading and might confuse readers. In the proposed capacitive kernel, inputs are capacitances while weights are voltage biases, which are contrary to what is depicted in these figures.*

Response: Thank you for your valuable suggestion. In our design, inputs are capacitances and weights are voltage biases in the proposed capacitive kernel, as you mentioned. In the original Fig. 1c, we intentionally positioned the capacitances (inputs) along the network lines to represent a matrix, which could maintain visual correspondence with the physical sensor array configuration shown in Fig. 1a. However, as you mentioned, this representation may cause confusion since artificial neural network architectures typically place inputs on the left and weights on the network lines. After thorough consideration, we have implemented your suggested modification by swapping these elements to align with standard conventions in the revised Fig. 1c. Supplementary Fig. 1 is kept unchanged to highlight the parallel behavior in the capacitive pressure sensor array.

We have modified **Figure 1c** and updated corresponding texts in the revised version to provide clearer representation for readers. (*Lines 113-115, Page 6*)

Figure 1c. Tactile artificial neural network constructed by the input capacitive pressure sensor vector C and predefined voltage matrix V .

Q6: From Fig. 2c and 2h, the response of the sensors to applied pressure seems to have 2 distinct regimes separated at around 1 kPa. Do the authors have any plausible explanation for this?

Response: Thank you for your comment. The nonlinear response behavior (distinct pressure-capacitance regimes) is a common phenomenon observed in capacitive sensors, which has been demonstrated in many previous literatures (*Nat Mater* 2010,9, 859–864; *Adv. Funct. Mater.*2020,30, 1903100). To further validate this phenomenon, we conducted comprehensive pressure-capacitance characterization across 30 sensor devices, as shown in Figure R13. These devices exhibit consistent pressure-capacitance responses. Although all devices have nearly linear response behaviors, two slightly distinct regimes separated at around 1 kPa were still observed. This phenomenon can be attributed to the increasing elastic resistance of the PVA/H₃PO₄ sensing film with increasing compression (*Adv. Funct. Mater.*2020,30, 1903100).

Figure R13 has been added as **Supplementary Figure 4a** in the revised version, with corresponding text description added. (*Lines 143-144, Page 8*)

Figure R13 Capacitance-pressure response of 30 capacitive sensor devices.

Q7: While in-sensor computing will undoubtedly reduce power consumption, I believe that the reported 4 orders of magnitude power advantage over a conventional system

is overestimated to some extent. The estimation for conventional system includes everything from ADC readout to periphery circuitries such as reference voltage generation, clocking, I/O ports, etc. However, for the in-sensor system, the static power of these circuitries (ADC to read the final output or voltage amps to drive subsequent analog operations, switching circuitries, clock for switching circuitries, input voltage generation, I/O ports, etc.) are not accounted for. I suggest the authors to refine their power calculation methods for a fairer comparison.

Response: Thank you for your constructive suggestion. According to your suggestion, the power consumption of peripheral circuitries such as switching circuitries, clock for switching circuitries and I/O ports in our system has been accounted into the total power consumption in the revised version (Our system does not include ADC and voltage amplifiers).

(1) The total power consumption of our capacitive in-sensor tactile computing system is calculated as following.

Power consumption of electrical switches: We used the TMUX1112 analog switch chips to implement the 19 electrical switches in the in-sensor computing system. The power consumption of these electrical switches consists of supply consumption, dynamic consumption and on-state consumption. We supplied a 3.3V power voltage to these switch chips. According to the user manual of the switch chip, the typical supply current is 5 nA, with a typical on-capacitance of 17 pF and an on-resistance of 3.7 Ω for each switch channel. Additionally, each of the switch channel was configured with a compliance current of 1 mA and a switching period of 61 ms (see Figure R8).

Supply consumption:

$$\begin{aligned} P_{supply} &= I_{supply} \times V_{DD} = 5 \text{ nA} \times 3.3 \text{ V} \\ &= 16.5 \text{ nW} \end{aligned}$$

Dynamic consumption:

$$\begin{aligned} P_{dynamic} &= C_{on} \times V_{DD}^2 \times f_{switching} = 17 \text{ pF} \times (3.3 \text{ V})^2 \times \frac{1}{61 \text{ ms}} \\ &= 3.09 \text{ nW} \end{aligned}$$

On-state consumption:

$$P_{on} = I_{on}^2 \times R_{on} = (1 \text{ mA})^2 \times 3.7 \Omega$$

$$= 3.7 \mu W$$

In the system, the 19 channels of the five TMUX1112 analog switch chips were used as 19 electrical switches.

$$P_{switches} = P_{supply} \times 5 + (P_{dynamic} + P_{on}) \times 19 = 82.5 \text{ nW} + (3.09 \text{ nW} + 3.7 \mu W) \times 19$$

$$\approx 70.4 \mu W$$

Power consumption of logic module for switching circuitries: To provide pulse control for the three sets of electrical switches in the system, we designed a switch control logic using Verilog and implemented it on an FPGA. The resultant power consumption of the switch control logic, includes specific I/O ports and the clock for switching circuitries, was evaluated on Vivado tool based on xczu15eg-ffvb1156-2-i device (Compared to the Quartus tool, the Vivado tool can provide more details on the power consumption of I/O ports, clock and logic). As shown in Figure R14, the power consumption of I/O ports and clock for switching circuitries is approximately 418.7 μ W.

Summary

Figure R14 Estimation of the power consumption for switch control logic.

Power consumption of the capacitive sensors and fixed capacitor C_0 : In noise reduction and edge detection tasks, the maximum power consumption was about 3.74 μ W and 2.58 μ W (see Methods).

Total power consumption:

In noise reduction task, the total power consumption of the system is:

$$P_{total} = P_{switches} + P_{IO/clock} + P_{sensors} = 70.4 \mu W + 418.7 \mu W + 3.74 \mu W$$

$$\approx 493 \mu W$$

In edge detection task, the total power consumption of the system is:

$$P_{total} = P_{switches} + P_{IO/clock} + P_{sensors} = 70.4 \mu W + 418.7 \mu W + 2.58 \mu W$$

$$\approx 492 \mu W$$

(2) **The total power consumption of conventional mixed electronic system** is calculated as following.

We used the Vivado tool to estimate the total power consumption of conventional mixed electronic system. As shown in Figure R15, the static power consumption represents the power consumed by the FPGA core, while the dynamic power consumption refers to the I/O ports, clock and logic. Therefore, the power consumption of the conventional mixed electronic system is approximately 11154 μ W. After considering the static power from peripheral circuitry, our system still exhibits a power consumption over 22 times lower than that of a conventional mixed electronic system.

Figure R15 Estimation of the average power consumption for the conventional mixed electronic system.

We have updated the power calculation methods of our system by accounting for peripheral circuitry effects in the revised version, which will provide a fairer comparison. Thank you again for your valuable suggestion. (Lines 28, Page 2; Lines 75, Page 4; Lines 304, Page 15; Lines 314-315, Page 16; Lines 328-329, Page 17)

Detailed responses to reviewers' comments

Reviewer #1:

Comments: *The authors have thought about the suggestions and comments. There is a suggestion. The authors listed similar research work only in the Supplementary Information. This makes it easy for readers to overlook relevant research. It is suggested that the authors should introduce these studies in the main text to help readers better understand the relevant progress.*

Response: We sincerely appreciate the reviewer' acceptance of our work and the valuable feedback throughout the review process. This suggestion has been adopted in the revised manuscript, as follows “In contrast, in-sensor computing paradigm utilizes individual self-adaptive sensors²⁷, multiple connected sensors¹³, or novel device structures^{28,29} to directly sense and simultaneously process sensory information, providing a more attractive solution with the relatively complete elimination of data conversion and transfer in the system”.

Reviewer #2:

Comments: *After a thorough review of the revised manuscript and its supplementary materials, I am satisfied that all of my questions and concerns have been fully addressed. Given the work's significant potential to advance integrated tactile sensing-computing systems, I strongly recommend this manuscript for publication in Nature Communications.*

Response: We sincerely appreciate your acceptance of our manuscript and your insightful suggestions.

Reviewer #3:

Comments: *The authors have addressed all of my comments and I do not have any other concerns. The manuscript is now greatly improved and can be accepted for publication.*

1. Page 16 line 314: The authors did not provide Supplementary Table 5. I suggest the authors to check again the manuscript for similar errors.

Response: We appreciate your thorough review. We confirm that the reference to Supplementary Table 5 should indeed be Supplementary Note 4, and we have verified the numbering of all figures and supplementary materials for accuracy.

Comments: *The authors envisioned the use of the proposed system to implement e-skin, which would benefit from high-density sensors for a more fine-grained sensing-processing in advanced applications. For the benefit of readers interested in this area, it would be good to add the brief discussion on comment #4 (possibility and feasibility of scaling down for a larger scale sensory integration) in the Supplementary Information.*

Response: According to your valuable suggestion, a brief discussion on comment #4 have been added into the in the Supplementary Note 3, as follow: “On the other hand, to enable large-area implementation of the capacitive in-sensor tactile computing array, the miniaturization of individual capacitive sensor units becomes necessary. However, this physical scaling may decrease both the absolute capacitance value and sensitivity of the sensor due to the reduced contact area. These reductions impose stringent requirements on the readout measurement system, demanding higher signal-to-noise ratios and lower parasitic capacitance. Consequently, a co-design optimization of capacitive sensing materials, sensor array and readout measurement system is essential for practical large-area implementations of capacitive tactile computing arrays.”

This article by Chen et al. reports on a tactile in-sensor computing system using an array of capacitive sensors. With proper periphery circuitries, the sensor array also functions as a switched-capacitor circuit and can perform analog MAC operation via charge sharing. Convolution-based preprocessing of tactile sensory signals such as noise filtering and edge detection was demonstrated.

The fabricated sensors seem to have very good uniformity. The idea of using capacitive array to perform MAC operations is not new, but incorporating capacitive sensor units into in-sensor MAC hardware is interesting. However, one major issue I have with this work is that the operation of the capacitive in-sensor array as demonstrated will face serious challenges when deployed in real-world applications (see comments #1-2), ultimately limiting its use to toy problems only. Several other issues are also listed below.

1. Based on the demonstration, the capacitance array acts as a convolution kernel. To process a single 2D tactile input, the array needs to be slid in both x and y directions. This does not seem like an elegant/logical/practical implementation: imagine having to physically slide the array under a cat's paw just to obtain a processed static tactile image.
2. The decay time C_0 is on the order of tens of seconds in Fig. 3d, but on the order of tens of milliseconds in Fig. 5d, e. Did the authors change any circuit parameters, or did they unintentionally mislabel the units? Also, if tens of milliseconds is needed to complete one MAC operation, the time needed to complete the preprocessing of the entire tactile image (see comment #1) would be unacceptably long. How should this issue be addressed?
3. How should the value of C_0 be determined? Is choosing a value such that $C_0 \gg \sum C_i$ the best strategy, perhaps in terms of SNR, power consumption, etc.? I believe that a smaller C_0 may increase the readout voltage range, which might be more desirable. Besides, with the current C_0 of 10 nF, $C_0 \gg \sum C_i$ does not really hold in most cases. Even the calculations in Supplementary Note 2 do not use this assumption. It would be less confusing if the authors simply omit this assumption.
4. How small can each capacitive sensor be physically scaled to while maintaining the appropriate absolute capacitance, range of capacitance change, sensitivity, resistance to noise, etc.? How significant is the increase in parasitic capacitances between adjacent cells upon scaling down and how do they affect computations? These are crucial when trying to achieve higher spatial resolution for a much wider range of applications.
5. Fig. 1c and Supplementary Fig. 1 is misleading and might confuse readers. In the proposed capacitive kernel, inputs are capacitances while weights are voltage biases, which are contrary to what is depicted in these figures.

6. From Fig. 2c and 2h, the response of the sensors to applied pressure seems to have 2 distinct regimes separated at around 1 kPa. Do the authors have any plausible explanation for this?
7. While in-sensor computing will undoubtedly reduce power consumption, I believe that the reported 4 orders of magnitude power advantage over a conventional system is overestimated to some extent. The estimation for conventional system includes everything from ADC readout to periphery circuitries such as reference voltage generation, clocking, I/O ports, etc. However, for the in-sensor system, the static power of these circuitries (ADC to read the final output or voltage amps to drive subsequent analog operations, switching circuitries, clock for switching circuitries, input voltage generation, I/O ports, etc.) are not accounted for. I suggest the authors to refine their power calculation methods for a fairer comparison.